# Secreted IgM modulates IL-10 expression in B cells

Shannon Eileen McGettigan [1], Lazaro Emilio Aira[1], Gaurav Kumar[2], Romain Ballet [3,4], Eugene C. Butcher [3,4], Nicole Baumgarth [5,6] & Gudrun F. Debes [1,7] ✉

IL-10[+] B cells are critical for immune homeostasis and restraining immune responses in infection, cancer, and inflammation; however, the signals that govern IL-10[+] B cell differentiation are ill-defined. Here we find that IL-10[+] B cells expand in mice lacking secreted IgM ((s)IgM[−/−]) up to 10-fold relative to wildtype (WT) among all major B cell and regulatory B cell subsets. The IL-10[+] B cell increase is polyclonal and presents within 24 hours of birth. In WT mice, sIgM is produced prenatally and limits the expansion of IL-10[+] B cells. Lack of the high affinity receptor for sIgM, FcμR, in B cells translates into an intermediate IL-10[+] B cell phenotype relative to WT or sIgM[−/−] mice. Our study thus shows that sIgM regulates IL-10 programming in B cells in part via B cell-expressed FcμR, thereby revealing a function of sIgM in regulating immune homeostasis.

Immune tolerance and maintenance of homeostasis are critical to restrain immune responses to infection and inflammation and to prevent autoimmunity. Several cell types maintain immune tolerance and homeostasis, including regulatory B and T lymphocytes (Breg and Treg). Secretion of the anti-inflammatory cytokine IL-10 is a key immunoregulatory mechanism. IL-10 is pleiotropic and protects the host from immunopathology during vigorous immune responses in infection and inflammation (reviewed in ref. [1]). Recently, the field has embraced the importance of B cells that secrete IL-10. Consistent with their ability to suppress immune responses, IL-10[+] Bregs are protective in settings of autoimmunity and graft rejection but are associated with poor outcomes in cancer and infection (reviewed in ref. [2]).

IL-10[+] B cells are heterogeneous, and no surface markers uniquely identify this population. However, IL-10[+] B cells are enriched within the CD1d[hi]CD5[+], T-cell immunoglobulin and mucin domain (TIM1)[+], CD9[+], transitional-2-marginal zone precursor (T2-MZP), B-1, marginal zone (MZ) B-cell populations, and more rarely in follicular (FO) B cells (reviewed in ref. [2]). IL-10 is also produced among IgA and IgM-secreting cells[3,4]. Development of IL-10[+] Bregs is incompletely understood but signaling through the B-cell receptor (BCR), Toll-Like Receptor (TLR) and CD40 induce optimal IL-10 competency[5], while additional signals such as BAFF, APRIL, and IL-21 enhance IL-10 programming (reviewed in ref. [2]). Once capable of IL-10 secretion, the IL-10 locus remains epigenetically open in B cells[6]. IL-10[+] B cells are expanded in young mice but remain at low frequencies (~1–3%) in the spleen of adult mice[5,7]. Similarly, in humans, blood-borne IL-10[+] B cells are enriched in cord blood and young children, peaking in middle childhood (5–11 y), and are retained at a low frequency (<2%) in healthy adults[8,9].

IgM is the first immunoglobulin isotype expressed developmentally. It can be membrane-bound as the BCR by alternative splicing, or secreted into local tissues or directly into blood circulation, typically as a large pentamer, ~950 kD in size[10]. Secreted IgM (sIgM) contributes to homeostasis including apoptotic cell and cell debris clearance (reviewed in refs. [10,11]). SIgM consists of both natural and adaptive IgM. Natural sIgM is produced constitutively

[1]Department of Microbiology and Immunology, Sidney Kimmel Medical College, Thomas Jefferson University, Philadelphia, PA 19107, USA. [2]Department of Cancer Biology, Sidney Kimmel Medical College, Thomas Jefferson University, Philadelphia, PA 19107, USA. [3]Palo Alto Veterans Institute for Research, Veterans Affairs Palo Alto Health Care System, Palo Alto, CA, USA. [4]Laboratory of Immunology and Vascular Biology, Department of Pathology, Stanford University School of Medicine, Stanford, CA, USA. [5]Center for Immunology and Infectious Diseases, Dept. Pathology, Microbiology & Immunology, University of California Davis, Davis, CA, USA. [6]Department of Molecular Microbiology and Immunology, Bloomberg School of Public Health, Johns Hopkins University, Baltimore, MD, USA. [7]Sidney Kimmel Cancer Center, Thomas Jefferson University, Philadelphia, PA 19107, USA. ✉e-mail: Gudrun.Debes@jefferson.edu

independent of microbes and exogenous antigens, largely by B-1 cell-derived antibody-secreting cells (ASC) (plasma cells and non-terminally differentiated cells) in bone marrow (BM) and spleen[12–15] as well as by human MZ B cells[16]. Whereas adaptive sIgM is generated in response to injury or infection[10]. SIgM binds to its high-affinity receptor FcµR and to Fcα/µR and CD22. FcµR is expressed on lymphocytes, specifically in humans on B, T, and natural killer (NK) cells and in mice largely on B cells[17]. Fcα/µR is bound by both IgM and IgA and is expressed by mature B cells, macrophages, and follicular dendritic cells but not immature B cells[18]. The immunoreceptor tyrosine inhibitory motif-containing CD22, a negative regulator of B-cell activation, recognizes sialic acid modifications on sIgM[19]. Additionally, sIgM attaches to polymeric Ig receptor (pIgR), which actively transports sIgM (and sIgA) through epithelia and allows for barrier tissue protection and homeostasis[20]. Immune complexes consisting of sIgM, antigen and complement bind cells via various complement receptors (reviewed in ref. 10). SIgM−/− mice retain surface IgM expression, class switch capacity, and ability to secrete other isotypes[21]. SIgM deficiency in humans and mice contributes to increased susceptibility to infection, which in mice is relieved by the administration of sIgM containing serum[22–25]. Unlike the regulatory

functions of IgG binding to FcγRIIB, the role of sIgM binding to its receptors on B cells is little explored.

In this study, we investigate the development of IL-10+ B cells. We find that in the absence of sIgM, mice have a significant increase in IL-10+ B cells among all major B-cell subsets. The IL-10+ B-cell expansion is polyclonal and due to the absence of sIgM in the environment and not the inability of the B cells to secrete IgM. Perinatal sIgM production is sufficient to restrict IL-10+ B cells, and reconstitution of sIgM production in sIgM-deficient mice significantly reduces IL-10+ B cells. SIgM acts in part through the B-cell-expressed FcµR to restrict IL-10 programming in B cells. Thus, our study reveals a role for sIgM in regulating the pool of IL-10+ B cells both early in life and during adulthood.

## Results

### Mice lacking secreted IgM have increased IL-10+ B cells

When revisiting the phenotype of sIgM−/− mice, we detected significantly increased serum IL-10 protein levels in naive sIgM−/− mice relative to wildtype (WT) controls ($p < 0.05$) (Fig. 1a). Similarly, *Il10* transcript expression in single-cell suspensions was significantly increased in spleen and subiliac lymph nodes (LN) ($p < 0.001$)

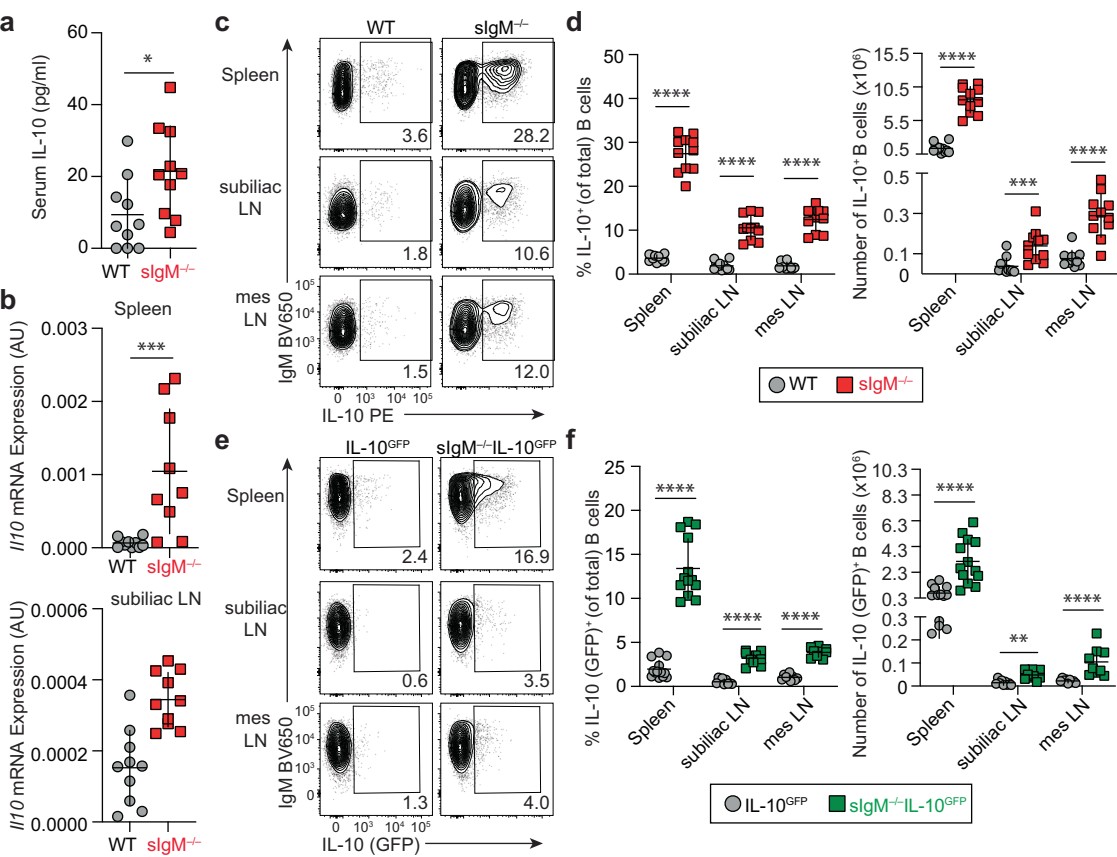

**Fig. 1 | Mice lacking secreted IgM have increased IL-10+ B cells. a** Serum concentration of IL-10 in naive WT and sIgM−/− mice by ELISA (WT $n = 10$, sIgM−/− $n = 10$, *$p = 0.0230$). **b** Relative expression of *Il10* transcript in cells isolated from the spleen and subiliac lymph node (LN) of WT and sIgM−/− mice at basal state determined by qPCR (Spleen: WT $n = 10$, sIgM−/− $n = 9$; ***$p = 0.0010$; subiliac LN: WT $n = 10$, sIgM−/− $n = 10$, ***$p = 0.0007$). **c** Representative flow cytometry staining of IL-10 protein in WT and sIgM−/− B cells (pregated on singlets, lymphocytes, Live/Dead− CD4−, CD19+) from the spleen, subiliac LN, and mesenteric (mes) LN after 4 h stimulation with phorbol 12-myristate 13-acetate/ionomycin/lipopolysaccharide. **d** Quantification of the percent and total number of IL-10+ B cells for each individual organ (WT $n = 11$, sIgM−/− $n = 11$; Left, percent IL-10+ B cells: spleen ****$p < 0.0001$, subiliac LN ****$p < 0.0001$, mes LN ****$p < 0.0001$; right, number of IL-10+ B cells:

spleen ****$p < 0.0001$, subiliac LN ***$p = 0.0003$, mes LN ****$p < 0.0001$). **e** Representative flow cytometry of IL-10 (GFP) expression in B cells directly ex vivo from IL-10GFP and sIgM−/−IL-10GFP reporter mice. **f** Quantification of the percent and total number IL-10+ B cells for each individual organ (Left, percent IL-10+ B cells: spleen (IL-10GFP $n = 13$, sIgM−/−IL-10GFP $n = 13$) ****$p < 0.0001$, subiliac LN (IL-10GFP $n = 9$, sIgM−/−IL-10GFP $n = 9$) ****$p < 0.0001$, and mes LN (IL-10GFP $n = 9$, sIgM−/−IL-10GFP $n = 9$) ****$p < 0.0001$; right, number of IL-10+ B cells: spleen (IL-10GFP $n = 9$, sIgM−/−IL-10GFP $n = 9$) ****$p < 0.0001$, subiliac LN (IL-10GFP $n = 9$, sIgM−/−IL-10GFP $n = 9$) **$p = 0.0012$, mes LN (IL-10GFP $n = 9$, sIgM−/−IL-10GFP $n = 9$) ****$p < 0.0001$). Data points indicate mean ± SD, and each symbol represents one mouse. *P* values were calculated using two-tailed Mann−Whitney *U* test (**a**, **b**, **d**, **f**). Data are pooled from 2 (**b**) or 3 (**a**, **d**, **f**) independent experiments. Source data are provided with this paper.

compared with WT mice (Fig. 1b). To reveal the cellular source of IL-10 we used a 4-h stimulation with phorbol 12-myristate 13-acetate/ionomycin/lipopolysaccharide (P/I/L), a method to detect IL-10 protein expression capability by flow cytometry[26]. We detected a 5.5- to 7.6-fold increase in percentage and a 3.7- to 6.3-fold increase in number of IL-10$^+$ B cells in the spleen, subiliac LN, mesenteric LN ($p < 0.001$) of naive sIgM$^{-/-}$ mice compared with WT controls (Fig. 1c, d; gating in Supplementary Fig. 1). On average, these differences resulted in 27% of B cells expressing IL-10 in spleens of sIgM$^{-/-}$ mice relative to 3% of IL-10 splenic B cells in WT mice (Fig. 1d). In vitro culture with and without anti-CD40 or LPS confirmed that splenic B cells from sIgM$^{-/-}$ mice secrete significantly more IL-10 than their WT counterparts (Supplementary Fig. 2a), consistent with increased IL-10 production by sIgM$^{-/-}$ B cells upon anti-CD40 stimulation using a different strain of sIgM$^{-/-}$ mice[27]. Analysis of IL-10$^{GFP}$-reporter mice (VertX[7]) crossed with sIgM$^{-/-}$ mice allowed us to evaluate IL-10$^+$ B cells in vivo without exogenous stimulation. During homeostasis, sIgM$^{-/-}$IL-10$^{GFP}$ mice had a significant 3.4- to 6.0-fold average increase in the number and percentage of IL-10$^+$ B cells in the spleen, subiliac LN, and mesenteric LN relative to IL-10$^{GFP}$ controls (Fig. 1e, f; frequencies, $p < 0.0001$; numbers $p < 0.01$). We evaluated other major leukocyte subsets, and we found only a minimal, but statistically significant, increase in the number of IL-10$^+$ splenic CD4 T cells (Supplementary Fig. 2b). This is unlikely due to a direct effect of sIgM on CD4$^+$ T cells as these cells lack sIgM-binding receptors, but it could potentially be due to the reported IL-10$^+$ B-cell-driven induction of IL-10$^+$ CD4$^+$ T cells (reviewed in ref. [2]). Consistent with previous reports[21,28], sIgM$^{-/-}$ mice had similar B-cell numbers and frequencies in lymph nodes and spleen, apart from a small but statistically significant decrease in splenic B-cell percentage and mesenteric LN B-cell numbers (Supplementary Fig. 2c, d). Thus, lack of sIgM led to the drastic accumulation of IL-10 expressing B cells without grossly affecting overall B-cell numbers.

### Lack of sIgM increases IL-10$^+$ B cells among all major B-cell subsets and Breg phenotypes

Next, we investigated if the increase in IL-10$^+$ B cells was restricted to a particular B-cell subset. We found that sIgM$^{-/-}$ mice had a significantly higher percentage and number of IL-10$^+$ B cells among all major splenic B-cell subsets relative to WT mice (Fig. 2a; $p < 0.001$). Specifically, the highest average fold increase in percent IL-10$^+$ B cells was in the FO (8.8-fold), followed by T2-MZP (7.0-fold), T2 (5.7-fold), T1 (4.1-fold), MZ (4.0-fold), and B-1 (1.2-fold) B-cell subsets (Fig. 2a; $p < 0.001$). The significant increase in IL-10$^+$ FO B-cell numbers ($p < 0.0001$) (Fig. 2a) existed despite a total decrease in FO B cells in sIgM$^{-/-}$ mice[21,28] (Supplementary Fig. 2e). Analyzing the BM B-cell fractions revealed that the increase in IL-10$^+$ B cells in sIgM$^{-/-}$ mice starts at the immature B-cell stage (Hardy fraction E; Fig. 2b) when the full BCR is first expressed on the B-cell surface.

Breg marker expression[5,29,30] was significantly increased on splenic B cells from sIgM$^{-/-}$ mice relative to WT, including TIM1$^+$ (2.0% in WT vs. 5.5% in sIgM$^{-/-}$ B cells; $p < 0.0001$), CD1d$^{hi}$CD5$^+$ (0.4% vs. 0.7%; $p < 0.05$), and CD9$^+$ (16% vs. 42%; $p < 0.0001$) (Fig. 2c). Importantly, Breg marker-positive B cells in sIgM$^{-/-}$ mice had a ~2–3-fold increase in percent IL-10 production relative to their WT counterparts: TIM1$^+$ (18% in WT vs. 56% in sIgM$^{-/-}$ B cells), CD1d$^{hi}$CD5$^+$ (34% vs. 68%), and CD9$^+$ (19% vs. 57%) (Fig. 2d; $p < 0.0001$). Finally, among IL-10$^+$ B cells in WT vs. sIgM$^{-/-}$ mice, we detected significantly increased expression of TIM1 (33% vs. 43%), and CD9 (79% vs. 92%), but a significantly decreased expression of CD1d$^{hi}$CD5$^+$ (5.9% vs. 2.5%) (Fig. 2e; $p < 0.01$). While the vast majority of both sIgM$^{-/-}$ and WT IL-10$^+$ B cells expressed CD9, only 19% of WT and 57% of sIgM$^{-/-}$ CD9$^+$ B cells were IL-10$^+$ (Fig. 2d, e).

ASCs, primarily IgM$^+$ and IgA$^+$, that also produce IL-10 are an important immunoregulatory population (reviewed in ref. [3,4]). We crossed the sIgM$^{-/-}$ strain with the Blimp-1$^{GFP}$ reporter mouse strain[31]. Blimp-1 (*Prdm1*) is a key transcription factor for ASC differentiation.

CD45$^+$CD3$^-$F4/80$^-$IgD$^-$ lymphocytes and IL-10$^+$ B cells from Blimp-1$^{GFP}$sIgM$^{-/-}$ and Blimp-1$^{GFP}$sIgM$^{+/-\ or\ +/+}$ mice did not differ in Blimp-1$^{GFP}$ expression (Fig. 2f), indicating the increase of IL-10$^+$ B cells in sIgM$^{-/-}$ mice was not due to arrested ASC development consistent with previous results demonstrating unperturbed IgG ASCs in sIgM$^{-/-}$ mice[28]. Together, these data indicate that lack of sIgM leads to IL-10 programming in all major B-cell and Breg subsets, independent of ASC differentiation.

### IL-10$^+$ B cells in sIgM$^{-/-}$ mice are polyclonal

To ascertain whether the drastic increase of IL-10$^+$ B cells in sIgM$^{-/-}$ mice was due to an oligoclonal outgrowth, we assessed the diversity of the BCR repertoire of sorted IL-10$^+$ (GFP$^+$) and IL-10$^-$ (GFP$^-$) splenic B cells from sIgM$^{-/-}$IL-10$^{GFP}$ mice. To account for differences in B-cell subset representation and their inherent differences in BCR diversity, we FACS-sorted B-1a, MZ, and FO populations (Fig. 3; gating strategy in Supplementary Fig. 4) whose BCR repertoires show restricted, intermediate, and high diversity, respectively[32,33]. We performed 10X single-cell VDJ repertoire analysis on the light chain sequences from the six populations. We were unable to analyze the heavy chain in most of the sequenced cells (see Methods for details). While there is less diversity in the light chain compared to the heavy chain[34], we successfully compared light chain diversity between the IL-10$^+$ and IL-10$^-$ B cells analyzing 2094-3467 cells per sample (Fig. 3a). Each sample yielded 804-1226 distinct clonotypes, and none of clonotypes in the IL-10$^+$ fraction exceeded 5.71% (Fig. 3a and Supplementary Data 1), indicating polyclonal IL-10$^+$ B-cell populations. Shannon entropy and Inverse Simpson diversity indices further confirmed that the IL-10$^+$ B cells were polyclonal in each analyzed B-cell subset. As expected, the IL-10$^+$ and IL-10$^-$ MZ and FO samples were more diverse than the B-1a subset, and the IL-10$^-$ FO was the most diverse sample (Fig. 3b). These data are consistent with our findings of increased IL-10$^+$ B cells among all B-cell subsets and collectively suggest that sIgM universally modulates IL-10 expression in B cells.

### The expansion of IL-10$^+$ B cells in the absence of sIgM appears perinatally

To determine when in ontogeny sIgM starts exerting its effects on IL-10 expression capacity in B cells, we analyzed sIgM$^{-/-}$ and WT mice as newborns <24 hours after birth, early in life on days 6 and 10, at weaning (day 22), and as adults, 8-18 weeks of age (Fig. 4a). Serum IgM levels increased as the WT mice matured, which was accompanied by an initial rise in splenic B-cell IL-10 expression from 0.9% to 8.5% between <24 h and day 6 of age; thereafter, the frequency of IL-10$^+$ B cells remained below 5% (Fig. 4a and Supplementary Fig. 5a–c). Akin to WT mice, sIgM$^{-/-}$ mice had an initial increase from 0.8% to 14.7% of IL-10 expression in splenic B cells between <24 h and day 6 of age. However, unlike the WT animals, the frequency of IL-10$^+$ B cells increased further to >20% as the mice matured (Fig. 4a). Splenic IL-10$^+$ B cells peaked on day 6 in WT mice but were exceeded significantly in age-matched sIgM$^{-/-}$ mice directly ex vivo and after a 4-h stimulation with P/I/L (Fig. 4b; $p < 0.001$). Specifically, the presence of sIgM in 6-day-old WT mice was associated with approximately 40% lower splenic and 30% lower liver B cells relative to age-matched sIgM$^{-/-}$ mice (Fig. 4b and Supplementary Fig. 5b). Notably, WT mice had detectable serum IgM (15–40 ng/mL) prenatally (~embryonic day 21) and at <24 hours of age (Fig. 4c). Comparing IL-10 expression in B cells without exogenous stimulation between WT and sIgM$^{-/-}$ mice, we found that sIgM$^{-/-}$ mice at <24 h after birth had significantly increased IL-10$^+$ B cells in the liver ($p < 0.01$), a difference not detectable after 4-h stimulation or in splenic B cells (Supplementary Fig. 5b, c). Furthermore, the significant increase of IL-10$^+$ B cells in sIgM$^{-/-}$ mice relative to WT mice was maintained in the spleen and liver at day 10 (spleen, $p < 0.001$; liver, $p < 0.05$) and day 22 in the spleen ($p < 0.0001$) and became more pronounced in splenic B cells of adult mice >8 weeks of age when IL-10 expression among sIgM$^{-/-}$ B cells increased to >20% (Fig. 1d, Supplementary Fig. 5; WT vs.

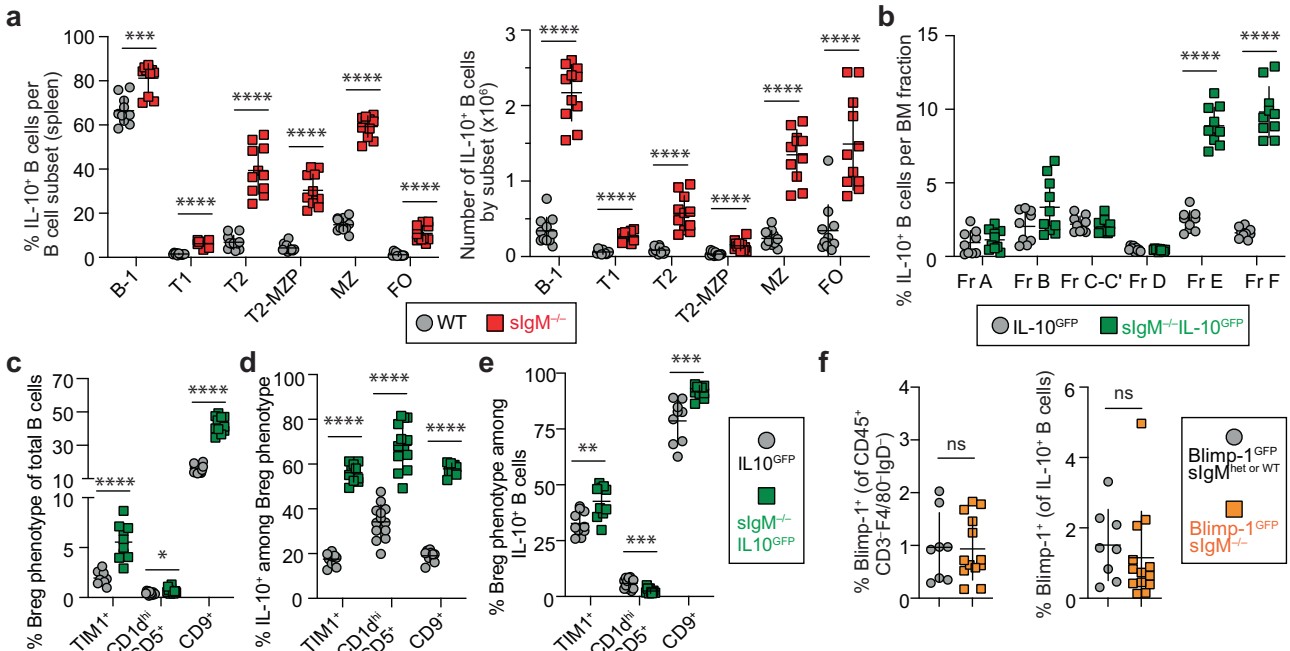

**Fig. 2 | Lack of secreted IgM increases IL-10$^+$ B cells among all major B-cell subsets and Breg phenotypes. a** Cells were stimulated with phorbol 12-myristate 13-acetate/ionomycin/lipopolysaccharide (P/I/L) for 4-h and gated on singlets, lymphocytes, live/dead (L/D) stain$^-$CD45$^+$CD4$^-$CD19$^+$, followed by markers specific to each B-cell subset: B-1: B220$^{lo/-}$CD43$^+$; and B-2: B220$^{hi}$CD43$^-$; B-2 cells were further subdivided as transitional-1 (T1): CD24$^{hi}$CD21$^{lo/-}$; transitional-2 (T2): CD24$^{hi}$CD21$^{int}$; transitional-2-marginal zone precursor (T2-MZP): CD24$^{hi}$CD21$^{hi}$CD23$^+$; marginal zone (MZ): CD24$^{hi}$CD21$^{hi}$CD23$^-$; and follicular (FO): CD24$^{int}$CD21$^{int}$CD23$^+$. The percent and total number of IL-10$^+$ B cells in each B-cell subset in the spleen of WT and sIgM$^{-/-}$ mice. (WT $n = 11$, sIgM$^{-/-}$ $n = 11$; left, percent IL-10$^+$ B cells: B1 ***$p = 0.0002$, T1 ****$p < 0.0001$, T2 ****$p < 0.0001$, T2-MZP ****$p < 0.0001$, MZ ****$p < 0.0001$, FO ****$p < 0.0001$; right, number of IL-10$^+$ B cells: B1 ****$p < 0.0001$, T1 ****$p < 0.0001$, T2 ****$p < 0.0001$, T2-MZP ****$p < 0.0001$, MZ ****$p < 0.0001$, FO ****$p < 0.0001$). **b** Bone marrow (BM) cells were assessed after 4-h stimulation with P/I/L for B-cell fractions according to the Hardy scheme. Cells were gated on singlets, Live/Dead (L/D) stain$^-$CD3$^-$F4/80$^-$NK1.1$^-$Gr-1$^-$B220$^+$. B220$^+$ BM cells were further subdivided into Hardy Fractions (Fr.): Fr. A: B220$^+$CD43$^+$IgM$^-$BP1$^-$CD24$^-$; Fr. B: B220$^+$CD43$^+$IgM$^-$BP1$^-$CD24$^+$; Fr. C-C': B220$^+$CD43$^+$IgM$^-$BP1$^+$CD24$^+$; Fr. D: B220$^+$CD43$^-$IgM$^-$IgD$^-$; Fr. E: B220$^+$CD43$^-$IgM$^+$IgD$^-$; and Fr. F: B220$^+$CD43$^-$IgM$^{+/-}$IgD$^+$. (IL-10$^{GFP}$ $n = 9$, sIgM$^{-/-}$IL-10$^{GFP}$ $n = 10$; percent IL-10$^+$ B cells: Fr. A $p = 0.3562$, Fr. B $p = 0.1823$, Fr. C-C'

$p = 0.3249$, Fr. D $p = 0.3555$, Fr. E ****$p < 0.0001$, Fr. F ****$p < 0.0001$. **c** Breg phenotype identified by expression of TIM1$^+$, CD1d$^{hi}$CD5$^+$, or CD9$^+$ among total splenic B cells without stimulation in sIgM$^{-/-}$IL-10$^{GFP}$ mice. (TIM1$^+$: IL-10$^{GFP}$ $n = 10$, sIgM$^{-/-}$IL-10$^{GFP}$ $n = 10$, ****$p < 0.0001$; CD1d$^{hi}$CD5$^+$: IL-10$^{GFP}$ $n = 13$, sIgM$^{-/-}$IL-10$^{GFP}$ $n = 13$, *$p < 0.0106$; CD9$^+$: IL-10$^{GFP}$ $n = 13$, sIgM$^{-/-}$IL-10$^{GFP}$ $n = 13$, ****$p < 0.0001$). **d, e** Splenocytes were stimulated for 4-h with P/I/L. **d** Frequency of IL-10$^+$ B cells among each Breg phenotype. (TIM1$^+$: IL-10$^{GFP}$ $n = 10$, sIgM$^{-/-}$IL-10$^{GFP}$ $n = 10$, ****$p < 0.0001$; CD1d$^{hi}$CD5$^+$: IL-10$^{GFP}$ $n = 13$, sIgM$^{-/-}$IL-10$^{GFP}$ $n = 13$, ****$p < 0.0001$; CD9$^+$: IL-10$^{GFP}$ $n = 9$, sIgM$^{-/-}$IL-10$^{GFP}$ $n = 9$, ****$p < 0.0001$). **e** Breg phenotype$^+$ cells among IL-10$^+$ B cells. (TIM1$^+$: IL-10$^{GFP}$ $n = 10$, sIgM$^{-/-}$IL-10$^{GFP}$ $n = 10$, **$p < 0.0089$; CD1d$^{hi}$CD5$^+$: IL-10$^{GFP}$ $n = 13$, sIgM$^{-/-}$IL-10$^{GFP}$ $n = 13$, ***$p < 0.0003$; CD9$^+$: IL-10$^{GFP}$ $n = 9$, sIgM$^{-/-}$IL-10$^{GFP}$ $n = 9$, ***$p < 0.0005$). **f** Plasma cell (PC) identity was assessed by flow cytometry directly ex vivo on L/D stain$^-$CD45$^-$CD3$^-$F4/80$^-$IgD$^-$ lymphocytes using sIgM$^{-/-}$ mice crossed with Blimp-1$^{GFP}$ reporter mice (left) and among IL-10$^+$ B cells after 4-h stimulation with P/I/L (right). (Blimp-1$^{GFP}$sIgM$^{het or WT}$ $n = 13$, Blimp-1$^{GFP}$sIgM$^{-/-}$ $n = 8$; Left, Percent Blimp-1$^+$ of PCs, $p = 0.8457$; Right, percent Blimp-1$^+$ of IL-10$^+$ B cells: $p = 0.2169$. Data points indicate mean ± SD, and each symbol represents one mouse. *P* values were calculated using two-tailed Mann–Whitney *U* test (**a**–**f**); not significant (ns). Data are pooled from 2 (**b**; **c**–**e**: TIM1$^+$/CD9$^+$) or 3 (**a**; **c**–**e**: CD1d$^{hi}$CD5$^+$; **f**) independent experiments. Source data are provided with this paper.

sIgM$^{-/-}$ $p < 0.0001$). These data demonstrate that the expanded IL-10$^+$ B-cell phenotype of sIgM manifests already at birth, and that under WT conditions, the presence of sIgM from birth limits the expansion of the IL-10$^+$ B-cell pool.

## SIgM produced within the mouse controls the pool of IL-10$^+$ B cells

We generated heterozygous sIgM$^{+/-}$IL-10$^{GFP}$ (IgH$^{b/a}$) mice by crossing sIgM$^{-/-}$IL-10$^{GFP}$ mice (IgH$^a$ allotype[21]) with sIgM$^{+/+}$IL-10$^{GFP}$ mice (IgH$^b$ allotype). Due to allelic exclusion, in a sIgM$^{+/-}$ mouse the IgM$^a$ B cells and IgM$^b$ B cells express the sIgM-knockout allele and the sIgM-wildtype allele, respectively. At 6 days after birth sIgM$^{+/-}$IL-10$^{GFP}$ (IgH$^{b/a}$) mice had serum IgM equivalent to age-matched sIgM$^{+/+}$IL-10$^{GFP}$ (IgH$^{b/b}$) mice, whereas sIgM$^{-/-}$IL-10$^{GFP}$ (IgH$^{a/a}$) mice had no detectable serum IgM, demonstrating that serum sIgM was produced within the sIgM$^{+/+}$ and sIgM$^{+/-}$ offspring and not the result of pregnancy or nursing by a sIgM-sufficient mother (Fig. 4d). To analyze B cells that express the sIgM-knockout allele and develop in an sIgM-sufficient environment and to exclude potential caging effects, we analyzed littermate

offspring from sIgM$^{+/-}$IL-10$^{GFP}$ (IgH$^{b/a}$) parents (Fig. 4e). Within a serum IgM sufficient sIgM$^{+/-}$IL-10$^{GFP}$ (IgH$^{b/a}$) 6-day old mouse, there was no difference in IL-10 (GFP) expression among B cells expressing the sIgM-wildtype allele (IgM$^b$) or the sIgM-knockout allele (IgM$^a$); there was also no difference in B-cell IL-10 expression in comparison to sIgM$^{+/+}$IL-10$^{GFP}$ (IgH$^b$) littermates (Fig. 4f; $p$ = not significant (ns)). In contrast, sIgM$^{-/-}$IL-10$^{GFP}$ (IgH$^a$) offspring had elevated frequencies of IL-10$^+$ B cells relative to their sIgM$^{+/+}$IL-10$^{GFP}$ (IgH$^b$) or sIgM$^{+/-}$IL-10$^{GFP}$ (IgH$^{b/a}$) littermates (Fig. 4f; $p < 0.01$). We extended this analysis to adult mice to confirm whether the presence of circulating sIgM continues to regulate IL-10 expression in B cells. Adult sIgM$^{+/-}$ (IgH$^{b/a}$) mice had serum IgM that trended lower than that of sIgM$^{+/+}$ (IgH$^{b/a}$) controls, a difference that did not attain statistical significance (Fig. 4g). IgH$^b$ and IgH$^a$ cells from sIgM$^{+/+}$ (IgH$^{b/a}$) control mice did not differ in the size of their IL-10$^+$ B-cell populations regardless of which wildtype IgH allotype was expressed (Fig. 4h). Similar to sIgM$^{+/+}$ (IgH$^{b/a}$) F1 control mice, in the sIgM$^{+/-}$ (IgH$^{b/a}$) F1 offspring, the sIgM-knockout (IgM$^a$) expressing B cells had an equal total IL-10$^+$ B-cell population relative to the sIgM-wildtype (IgM$^b$) expressing B cells in an sIgM-sufficient mouse (Fig. 4h).

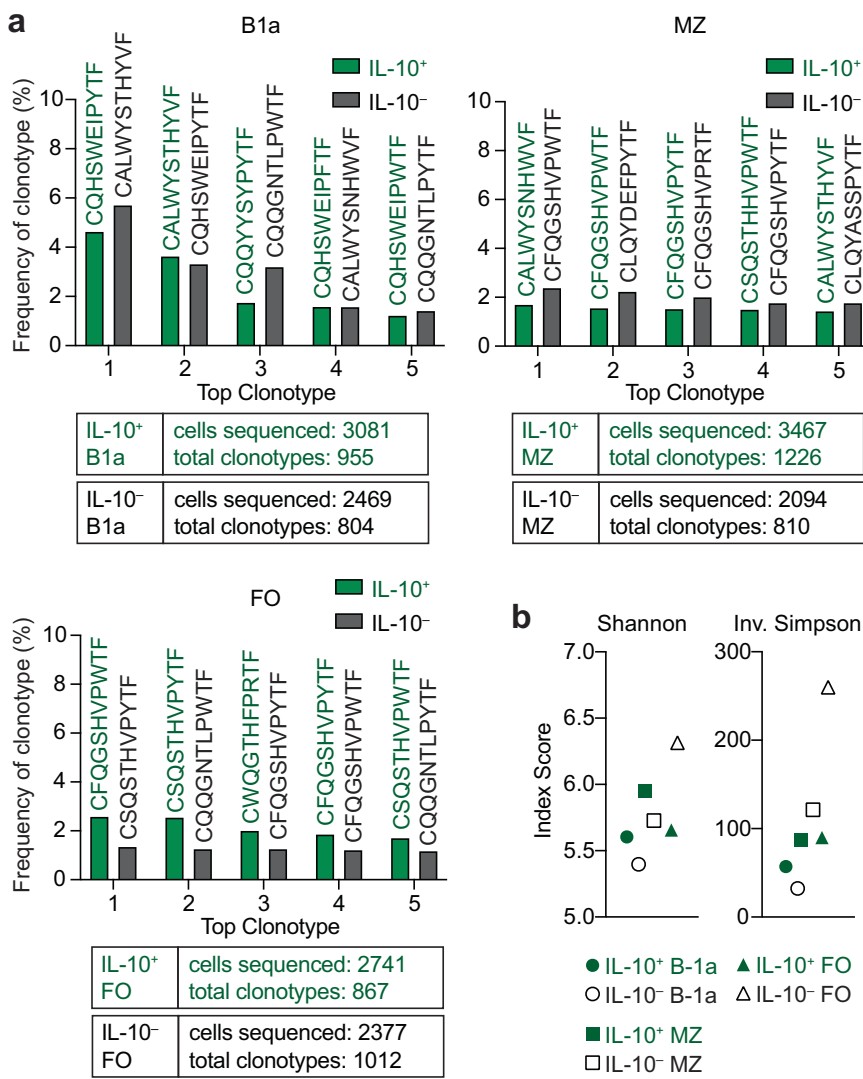

**Fig. 3 | IL-10⁺ B cells from sIgM⁻/⁻ mice are polyclonal.** Single-cell light chain clonotype analysis using the 10X Genomics platform of FACS-sorted IL-10⁺ (green) and IL-10⁻ (black) B-1a, marginal zone (MZ), and follicular (FO) B cells from sIgM⁻/⁻IL-10^GFP mouse after 4-h stimulation with phorbol 12-myristate 13-acetate/ionomycin/lipopolysaccharide, gated as in Supplementary Fig. 4. *Il10* transcript-expressing cells were removed from GFP⁻ sorted populations. Data were analyzed with Cell Ranger VDJ Scripts and plotted using R. **a** The percentage of top five individual clonotypes from each sample is shown with the corresponding CDR3 sequence listed on top. The total cells analyzed and total clonotypes identified are indicated below graphs. Complete information on VJ gene usage and nucleotide sequences found in Supplementary Data 1. **b** Diversity indexes (Shannon and Inverse Simpson) were assessed for the 6 samples using scRepertoire.

Transfer of WT peritoneal cavity (PerC) cells gives rise to IgM secreting cells in BM and spleen of recipient mice[35]. We used this system to partially restore IgM production, yielding 4.4 µg/ml serum sIgM, the equivalent of ~3% age-matched WT serum after reconstitution of 4.5–7–week–old sIgM⁻/⁻ mice with WT PerC cells (Fig. 4i). This treatment significantly reduced the frequency of IL-10⁺ cells among total B cells, B-1, MZ, and FO B cells by 15–40% relative to untreated sIgM⁻/⁻ mice ($p < 0.01$; Fig. 4i). In contrast, reconstitution of sIgM⁻/⁻ mice with sIgM⁻/⁻ PerC cells did not induce serum IgM or alter frequencies of IL-10⁺ B cells (Fig. 4i). We conclude that the expansion of IL-10⁺ B cells in sIgM-deficient mice are not the result of the inability of B cells to secrete IgM, instead, the presence of sIgM in the environment regulates IL-10 expression in B cells.

### Lack of FcµR partially phenocopies the IL-10-B-cell phenotype of sIgM-deficient mice

To determine if the IL-10⁺ B-cell expansion in the absence of sIgM is mediated by sIgM-binding receptors expressed on B cells, we analyzed IL-10 expression in B cells from CD22 and FcµR knockout mice.

Unlike a previous report[5], we did not detect differences in splenic IL-10⁺ B cells in CD22⁻/⁻ compared to WT mice (Fig. 5a). In contrast, IL-10⁺ B cells were increased in both total and B-cell-specific FcµR-deficient mice, *Cmv*^cre/WT*Fcmr*^fl/fl (FcµR⁻/⁻) and *CD19*^cre/WT*Fcmr*^fl/fl (FcµR^B cell⁻/⁻)[36] respectively (Fig. 5b). The IL-10⁺ B-cell phenotype in FcµR⁻/⁻ mice was intermediate relative to WT and sIgM⁻/⁻ mice, and there was no significant difference in IL-10 expression among B cells from FcµR⁻/⁻ and FcµR^B cell⁻/⁻ mice (Fig. 5b). These data illustrate a role for B-cell-expressed FcµR as well as alternative mechanisms in sIgM-mediated regulation of IL-10 expression programming in B cells.

### Discussion

Here, we discovered an unexpected relationship between sIgM and IL-10 programming in B cells. We found that a lack of sIgM leads to enhanced IL-10 imprinting in all major B-cell and Breg subsets. This included both B cells known for IL-10 expression and B-cell subsets rarely associated with IL-10 production, such as FO B cells, introducing sIgM as a universal regulator of IL-10 programming in

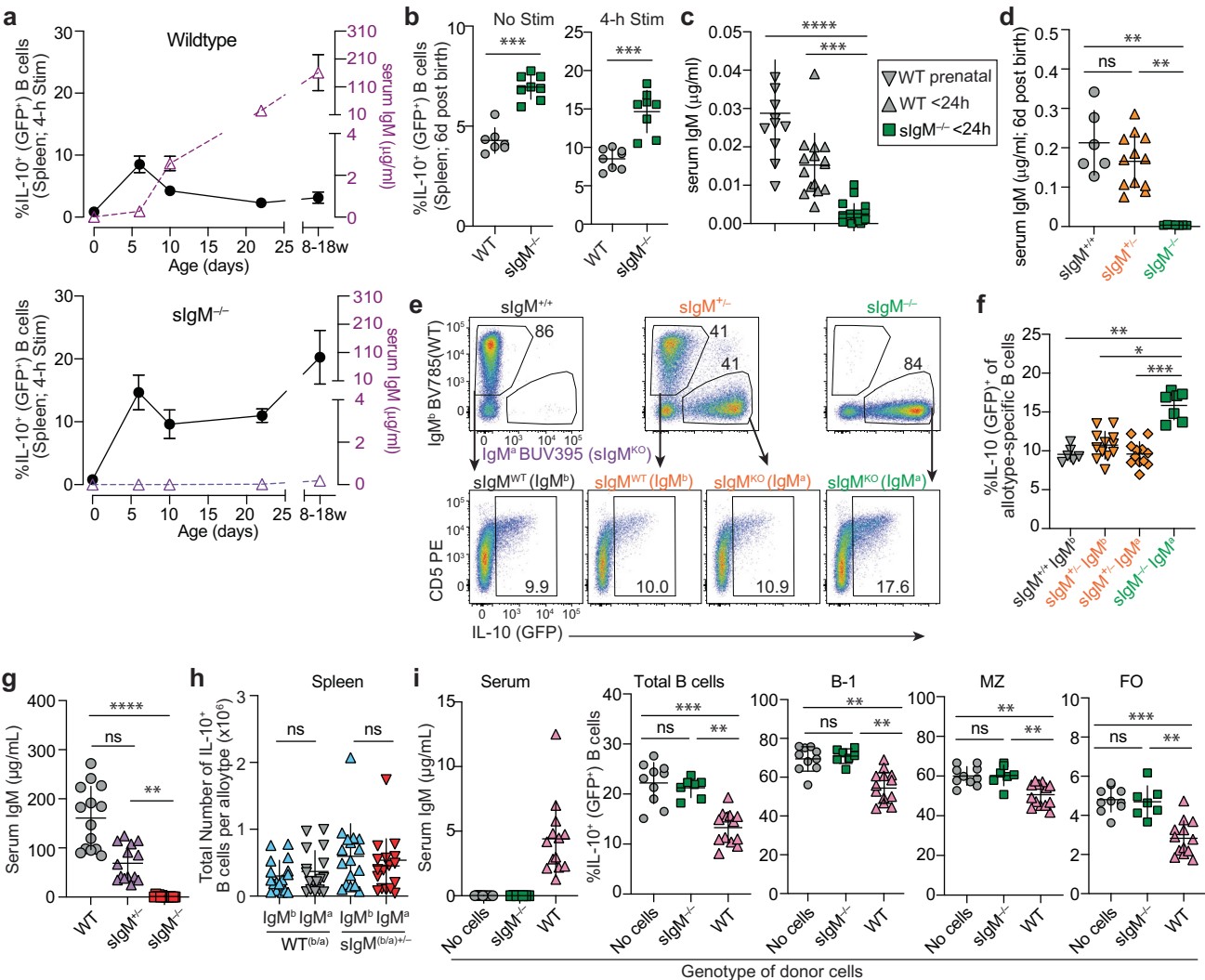

**Fig. 4 | SIgM is produced perinatally and limits the expansion of IL-10⁺ B cells.**
**a** Flow cytometric analysis of splenic IL-10⁺ B cells at <24 h post birth, d6, d10, d22, and adult mice (8–18 weeks) (left y axis; black) after 4-h stim with phorbol 12-myristate 13-acetate/ionomycin/lipopolysaccharide (P/I/L) and serum IgM by ELISA (right y axis; purple) of WT and sIgM⁻/⁻ mice. WT serum (<24 h $n = 15$, d6 $n = 8$, d10 $n = 7$, d22 $n = 10$, adult $n = 14$); WT IL-10^GFP percent IL-10⁺ B cells (<24 h $n = 7$, d6 $n = 8$, d10 $n = 7$, d22 $n = 10$, adult $n = 18$); sIgM⁻/⁻ serum (<24 h $n = 19$, d6 $n = 8$, d10 $n = 8$, d22 $n = 10$, adult $n = 9$); sIgM⁻/⁻IL-10^GFP percent IL-10⁺ B cells (<24h $n = 8$, d6 $n = 8$, d10 $n = 8$, d22 $n = 10$, adult $n = 18$). **b** Splenic IL-10⁺ B cells from WT and sIgM⁻/⁻ mice at basal state (left, IL-10^GFP $n = 7$, sIgM⁻/⁻ IL-10^GFP $n = 8$, ***$p = 0.0003$) and after 4-h P/I/L stimulation (right, IL-10^GFP $n = 8$, sIgM⁻/⁻ IL-10^GFP $n = 8$, ***$p = 0.0002$). **c** Serum IgM levels in prenatal WT (-E20-21, $n = 11$) and <24 h post birth in WT ($n = 15$) and sIgM⁻/⁻ ($n = 19$) mice (WT prenatal vs. WT < 24 h $p = 0.2131$, WT prenatal vs. sIgM⁻/⁻ < 24 h ****$p < 0.0001$, WT < 24 h vs. sIgM⁻/⁻ < 24 h ***$p < 0.0003$). **d–f** Analysis of sIgM⁺/⁺IL-10^GFP, sIgM⁺/⁻IL-10^GFP and sIgM⁻/⁻ IL-10^GFP littermates. **d** Serum IgM from day 6 post birth (sIgM⁺/⁺IL-10^GFP $n = 6$, sIgM⁺/⁻IL-10^GFP $n = 12$ and sIgM⁻/⁻IL-10^GFP $n = 7$; +/+ vs. +/− $p > 0.9999$, +/+ vs. −/− **$p > 0.0016$, +/− vs. −/− **$p > 0.0026$). **e** Representative flow cytometric analysis of splenic B cells gated on IgMᵃ (sIgM-knockout allele) and IgMᵇ (sIgM-wildtype allele) and assessed for IL-10 (GFP) expression after 4-h P/I/L stimulation. **f** Frequency of IL-10⁺ B cells based on allotype expressed (sIgM⁺/⁺IL-10^GFP $n = 6$, sIgM⁺/⁻IL-10^GFP $n = 12$ and sIgM⁻/⁻IL-10^GFP $n = 7$; sIgM⁺/⁺IgMᵇ vs. sIgM⁺/⁻IgMᵇ $p > 0.9999$, sIgM⁺/⁺IgMᵇ vs. sIgM⁺/⁻IgMᵃ $p > 0.9999$, sIgM⁺/⁺IgMᵇ vs. sIgM⁻/⁻IgMᵃ **$p > 0.0033$, sIgM⁺/⁻IgMᵇ vs. sIgM⁺/⁻IgMᵃ $p > 0.9999$, sIgM⁺/⁻IgMᵇ vs. sIgM⁻/⁻IgMᵃ

*$p = 0.0297$, sIgM⁺/⁻IgMᵃ vs. sIgM⁻/⁻IgMᵃ ***$p = 0.0004$). **g, h** Analysis of adult F1 offspring of WT.IgHᵃ/ᵃ x WT.IgHᵇ/ᵇ and sIgM⁻/⁻.IgHᵃ/ᵃ x WT.IgHᵇ/ᵇ mice. **g** Serum IgM concentration determined by ELISA in adult WT ($n = 14$), sIgM⁺/⁻ ($n = 14$), and sIgM⁻/⁻ ($n = 14$) mice (WT vs. Het $p = 0.0678$, WT vs KO ****$p < 0.0001$, Het vs. KO **$p = 0.0021$). **h** Flow cytometric analysis of splenic B cells, gated on IgMᵃ and IgMᵇ and assessed for IL-10 expression after 4-h P/L stimulation. Total number of IL-10⁺ B cells per allotype in spleen (WT.IgHᵇ/ᵃ $n = 17$, IgMᵇ⁺ vs. IgMᵃ⁺ $p = 0.4958$; sIgM⁺/⁻(ᵇ/ᵃ) $n = 17$, IgMᵇ⁺ vs. IgMᵃ⁺ $p = 0.2415$). **i** sIgM⁻/⁻IL-10^GFP mice 5–8 weeks after adoptive transfer of WT or sIgM⁻/⁻ PerC cells. Serum IgM by ELISA (Donor cell genotype: no cells, $n = 10$, sIgM⁻/⁻ $n = 7$, WT $n = 13$) and flow cytometric analysis of IL-10 (GFP) in splenic B-cell subsets after 4-h P/I/L stimulation (Donor cell genotype: no cells, $n = 10$, sIgM⁻/⁻ $n = 7$, WT $n = 13$. % IL-10⁺ of Total B cells: No cells vs. sIgM⁻/⁻ $p > 0.9999$, No cells vs. WT **$p = 0.0002$, sIgM⁻/⁻ vs. WT **$p = 0.0068$; percent IL-10⁺ of B-1 B cells: No cells vs. sIgM⁻/⁻ $p > 0.9999$, No cells vs. WT **$p = 0.0015$, sIgM⁻/⁻ vs. WT **$p = 0.0023$; percent IL-10⁺ of MZ B cells: No cells vs. sIgM⁻/⁻ $p > 0.9999$, No cells vs. WT **$p = 0.0032$, sIgM⁻/⁻ vs. WT **$p = 0.0049$; percent IL-10⁺ of FO B cells: No cells vs. sIgM⁻/⁻ $p > 0.9999$, No cells vs. WT ***$p = 0.0007$, sIgM⁻/⁻ vs. WT **$p = 0.0088$). **a–i** Data points indicate mean ± SD, and each symbol represents one mouse. *P* values were calculated using two-tailed Mann–Whitney U test (**b, h**) and Kruskal–Wallis test with Dunn's multiple comparisons test (**c, d, f, g, i**); not significant (ns). Data are pooled from 2 (**d, f, h**), 3 (**c, g, i**), 4 (**b**) or 6 (**a**) independent experiments. Source data are provided with this paper.

B cells. The inverse correlation between sIgM levels and IL-10⁺ B cells in ontogeny, together with the finding that restoration of even small levels of sIgM production reduces IL-10⁺ B cells in sIgM⁻/⁻ mice, confirm that sIgM governs the size of the IL-10⁺ B-cell pool.

For host defense, neonates rely on maternal IgG and IgA that are passively provided to offspring during pregnancy via circulation (IgG) and/or via mother's milk (IgG and IgA). IgG is selectively transferred into serum by the FcRn, and maternal IgA coats the respiratory and gut

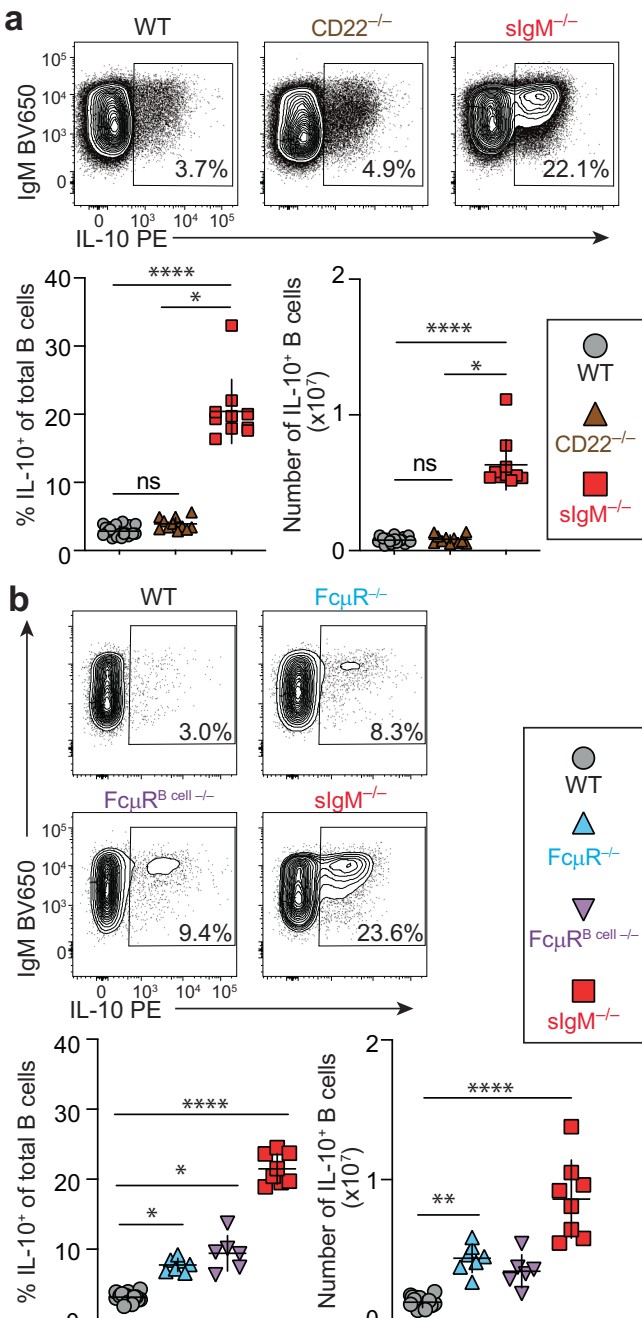

**Fig. 5 | Lack of FcμR partially phenocopies the IL-10⁺ B-cell phenotype of sIgM⁻/⁻ mice. a, b** Flow cytometric analysis of IL-10 expression after 4-h stimulation with phorbol 12-myristate 13-acetate/ionomycin/lipopolysaccharide in splenic B cells. **a** WT ($n = 21$), CD22⁻/⁻ ($n = 11$), and sIgM⁻/⁻ ($n = 10$), WT vs. CD22⁻/⁻ $p = 0.0608$, WT vs. sIgM⁻/⁻ ****$p < 0.0001$, CD22⁻/⁻ vs. sIgM⁻/⁻ *$p < 0.0263$. **b** WT ($n = 16$), FcμR⁻/⁻ ($Cmu^{cre}Fcmr^{-/-}$) ($n = 7$), FcμR^{B cell−/−} ($CD19^{cre} Fcmr^{fl/fl}$) ($n = 6$), and sIgM⁻/⁻ mice ($n = 8$) (WT vs. FcμR⁻/⁻ *$p = 0.0462$, WT vs. FcμR^{B cell−/−} *$p = 0.0109$, WT vs. sIgM⁻/⁻ ****$p < 0.0001$, FcμR⁻/⁻ vs. FcμR^{B cell−/−} $p > 0.99999$, FcμR⁻/⁻ vs. sIgM⁻/⁻ $p = 0.1994$, FcμR^{B cell−/−} vs. sIgM⁻/⁻ $p = 0.7847$). Data points indicate mean ± SD, and each symbol represents one mouse. $P$ values were calculated using Kruskal–Wallis test with Dunn's multiple comparisons test; not significant (ns). Data are pooled from 2 independent experiments. Source data are provided with this paper.

passes neither the placental barrier nor the murine neonatal gut, which has increased permeability to molecules up to 150 kD until 2-3 weeks post birth[10,39]. We demonstrate sIgM in the circulation of WT mice pre-birth and in the neonatal period, a finding in line with confirmed pre-natal sIgM production in other species, including humans[40] and rats[41] likely offsetting the lack of maternally provided sIgM. We report that sIgM present in sIgM⁺/⁻ and WT mice prevents enhanced expansion of IL-10⁺ B cells in both neonates and adult mice. Birth and early feeding and nursing represent significant events of exposure to novel microbes, colonization and antigens, and mechanisms to minimize overactivation of the newborn immune system must be in place to prevent immunopathology. The reliance of the developing organism on its own sIgM without the introduction of maternally derived sIgM calls for an alternative mechanism to fine-tune early immune tolerance to novel antigens and microbial products. IL-10 secretion by B cells may serve to maintain tolerance early in life while sIgM levels are building, as B-1 derived plasma cells, a major source of sIgM, are not generated in mice until 3 weeks of age[15]. This notion is further supported by the higher frequency of IL-10⁺ B cells in neonatal compared to adult WT mice (Fig. 4 and refs. 5,42), consistent with the enrichment of blood-borne IL-10⁺ B cells in cord blood and young children (peaking between 5-11 y) compared to <2% IL-10⁺ B cells in healthy teens and adults[8,9]. In addition, the transcription factor, aryl hydro-carbon receptor (AhR), binds upstream to the transcription start site of the *Il10* locus and promotes IL-10⁺ Breg differentiation in B cells[43]. AhR ligands include dietary compounds and gut microbe-derived trypto-phan metabolites[44]. Thus, post-natal microbial colonization and initiation of feeding should provide a sharp increase in AhR ligands that may have contributed to the early expansion of IL-10⁺ B cells in our experiments.

Our data in sIgM⁺/⁻ mice demonstrate that sIgM in the local tissue environment and not the inability of a B-cell to secrete IgM regulates its IL-10 programming because sIgM-deficient B cells that develop in a sIgM-replete environment do not display enhanced IL-10 program-ming. Tissue sIgM levels may also influence IL-10 B-cell programming in situ as tissue sIgM levels depend on local production or diffusion. Due to its size (~950 kD), sIgM diffuses poorly into tissues as most blood vessel endothelia only allow for paracellular transport of mole-cules up to 70 kD[45] and passage into tissues would require transcellular transport via a receptor, such as endothelial expressed pIgR[46]. How-ever, this has never been demonstrated for pentameric sIgM[45], and models of tissue ischemia and reperfusion demonstrate that infiltra-tion of self-reactive natural IgM and complement exacerbate tissue injury following endothelial damage and leakage[47]. Thus, tight reg-ulation of sIgM in tissues through controlled local production may serve homeostasis. Indeed, sIgM is constitutively produced in both lymphoid and non-lymphoid tissues, such as skin, lung, liver, and intestine, and IgM-ASCs accumulate in inflamed tissues[14,48,49]. Inter-estingly, skin-localized IgM-ASCs depend on APRIL or BAFF whose cutaneous expression is increased in chronic inflammation[49–51]. Thus, targeting ASC survival factors could be a tool to affect local sIgM production and IL-10 programming in B cells at the site of inflamma-tion or infection. Another mechanism to regulate local sIgM produc-tion may occur through regulatory T cells, which were reported to negatively regulate natural IgM production in the BM[52].

Natural sIgM is involved in the clearance of apoptotic cells[10], and apoptotic cells induce IL-10⁺ regulatory B cells that are protective in inflammation models such as arthritis and experimental autoimmune encephalitis[53,54]. Thus, an increase in IL-10 expressing B cells in sIgM⁻/⁻ mice could reflect an outgrowth of B-cell clones that recognize sIgM targets. Our data showed that IL-10⁺ and IL-10⁻ sIgM⁻/⁻ B cells have comparable diversity and overlapping clonotypes based on light chain usage. FO B cells, which have the most diverse BCR repertoire relative to MZ and B-1a B cells, had only minimally restricted BCR diversity among the IL-10⁺ B cells when compared to their IL-10⁻ counterparts,

mucosal surfaces while early humoral responses develop in the neonate[37,38]. Our study showed that sIgM from sIgM-sufficient mothers does not cause detectable serum IgM in nursing sIgM-deficient off-spring. This is consistent with the notion that sIgM, ~950 kD in size,

arguing against an oligoclonal outgrowth of IL-10[+] B-cell clones. Our data also confirm studies by Maseda et al. showing that WT IL-10[+] B cells have a diverse BCR repertoire[55].

FcμR is expressed on all B-cell subsets, in line with our data showing an increase in IL-10 expression among all major B-cell subsets in sIgM[−/−] mice. Strikingly, FcμR expression is highest on the T2-MZP and FO subsets[56], consistent with our data demonstrating the highest relative increase in IL-10 expression in FO and T2-MZP B cells in sIgM[−/−] mice. Thus, the data support a major role for B-cell-expressed FcμR in sIgM-driven regulation of IL-10 competency in B cells. Five strains of FcμR[−/−] mice have been generated using different gene-targeting strategies, resulting in varied phenotypes (reviewed in detail in[17]). Our results, using the Baumgarth FcμR[−/−] mouse strain[36], are consistent with a study showing increased IL-10[+] B cells in influenza virus-infected FcμR[−/−] mice (Lee strain[57]). However, the Ohno FcμR[−/−] strain exhibits reduced IL-10 production by B cells in vitro[58]. Only exon 4 was targeted in our FcμR[−/−] strain[36], whereas all other strains targeted multiple *Fcmr* exons[17] and could account for discrepancies in IL-10 phenotypes. Of note, the mouse *Fcmr* and *Il10* loci are in close proximity (-139 kb). Yanaba et al.[5] showed a ~ 2.5-fold increase in IL-10[+] B cells in CD22[−/−] mice relative to WT mice, however, we were unable to recapitulate the results using a different strain of CD22[−/−] mice[59]. Besides mouse strain differences, possible reasons for the divergent results include differences in age (we analyzed slightly younger mice), as well as variations in animal housing or experimental protocols, such as in vitro stimulation conditions. While we cannot formally rule out a role for CD22 in regulating IL-10 programming in B cells our data suggest that it is at least redundant with other sIgM-binding receptors. Possible alternative sIgM-binding receptors on B cells that may regulate IL-10 programming include Fcα/μR, pIgR or complement receptors 1 and 2 (CR1 and CR2). However, Fcα/μR is not expressed on all B-cell subsets[60], pIgR expression is largely restricted to epithelial cells[61], and CR1 and CR2 are widely expressed by B-cell subsets but also bind complement independent of sIgM[62]. Therefore, it is likely that sIgM-binding to multiple B-cell-expressed receptors is required for the regulation of IL-10 programming in B cells. Pentameric sIgM also serves as a carrier for apoptosis inhibitor of macrophages (AIM/CD5L), an effector protein with functions such as anti-apoptotic activity, biological debris removal and inflammatory response modulation[63,64]. Alternatively, sIgM may regulate IL-10 programming in B cells indirectly via sIgM-binding receptors expressed on non-B cells, and future studies are required to address the potential roles for alternative IgM binding receptors or indirect regulation through other cell types.

Imprinting of IL-10 expression capacity in B cells requires activation of a combination of receptors: BCR, TLR, and CD40[2,5]. We are unaware of data that support a role for sIgM in regulating TLR or CD40 signaling. In contrast, B cells from sIgM[−/−] mice have increased BCR signaling, and provision of polyclonal sIgM or treatment with ibrutinib, which inhibits Btk downstream of the BCR, restores physiological BCR signaling strength[65,66]. Our finding that lack of sIgM boosts IL-10 production in Hardy fraction E (immature B cells) when the fully assembled BCR is expressed on the cell surface support the notion that sIgM inhibits IL-10 programming via affecting BCR activity. Recent structural data on FcμR binding of sIgM also support a role in modulating BCR signaling[67–69]. One pentameric sIgM has 8 potential binding sites for FcμR with demonstrated binding of 4 FcμR molecules, and one BCR IgM binds two FcμR molecules[67–69]. While the signaling capacities of FcμR are not fully elucidated, there is potential for clustering of FcμR, phosphorylation of conserved tyrosine/serine residues in the cytoplasmic tail of FcμR, or interactions via putative adaptor proteins which may potentially modulate BCR signaling[17,69] and therefore affect IL-10 programming in B cells. The intermediate effect we observed in FcμR[−/−] compared to WT and sIgM[−/−] mice suggests that there are additional mechanisms regulating BCR signaling and IL-10 programming. For example, sIgM could indirectly affect BCR signaling

by acting as a decoy receptor that restricts antigen binding to the BCR[66]. Deficiency in sIgM impairs pathogen clearance following infection in mice and humans[22–25], and increased pathogen load will lead to the accumulation of both antigens and TLR ligands that may contribute to IL-10 production in B cells and further impede pathogen clearance. Interestingly, humans with selective IgM deficiency have a small but significant increase in the frequency of B cells with a Breg phenotype[70]; however, IL-10 expression has yet to be addressed.

Taken together, induction of IL-10 production by B cells may serve as a self-restraint mechanism of the immune system and limit collateral damage during overactivation akin to effector T cells that produce IL-10 during overt inflammation even at the risk of impaired pathogen clearance (reviewed in refs. 71,72). We propose that the increase in IL-10 production by B cells functions as a counter mechanism to limit inflammation in an environment in which homeostatic functions of sIgM are missing temporally or spatially, such as early in life or within tissues, or permanently as in patients with selective IgM deficiency. Future studies should address the role of enhanced B-cell IL-10 expression in restraining B-cell and other immune responses in the absence of sIgM. Modulating the function of IL-10[+] B cells by targeting sIgM in tissue sites could provide future avenues to treat a wide range of diseases in which IL-10[+] B cells play an important role, ranging from inflammation to chronic infection and cancer.

## Methods

### Mice and tissues

All mice (*Mus musculus*) were on a C57BL/6 background and provided food and water ad libitum under specific pathogen-free housing conditions at 68–72°F, 30–70% humidity with a 12/12 light/dark cycle. Experimental mice and control mice were bred separately except for littermate studies in newborns and adoptive transfer studies in which mice were co-housed. Mice were euthanized by carbon dioxide inhalation using a euthanasia station with a dedicated chamber, compressed gas cylinder, and regulated flow rate combined with subsequent cervical dislocation of apparent dead animals. All procedures were approved by the Institutional Care and Use Committees of Thomas Jefferson University (TJU), the VA Palo Alto Health Care System, and the University of California at Davis. Sex was not considered in the study design as we did not see a sex bias in initial phenotype results, but we age- and sex-matched groups of male and female mice in all experiments. Mice were 8–18 weeks of age in all experiments except where specifically noted (newborn studies) and bred in-house. C57BL/6 (Stock number, 000664), B6.SJL (002014), IgHa congenic (001317), and IL-10[GFP](VertX) (014530) mouse strains were obtained from The Jackson Laboratory. sIgM[−/−] mice[21] were obtained from Frances Lund (University of Alabama at Birmingham) and backcrossed for n > 10 generations with C57BL/6 mice. Blimp-1[GFP] mice[31] and CD22[−/−] mice[59] were provided by Stephen Nutt (Walter and Eliza Hall Institute of Medical Research) and James Paulson (Scripps Institute), respectively. *Cmu*[cre]*Fcmr*[−/−], *CD19*[cre/+]*Fcmr*[fl/fl] mice were generated previously[36].

Spleens and subiliac lymph nodes from *Cmu*[cre]*Fcmr*[−/−], *CD19*[cre/+]*Fcmr*[fl/fl], and CD22[−/−] mice, as well as sex- and age-matched WT controls were harvested intact in complete RPMI (RPMI 1640 with 25 mM HEPES, 1× Glutamax, 100 U/mL penicillin, 100 μg/mL streptomycin, 10% serum, 50 μM 2-mercaptoethanol) and shipped overnight on wet ice from the UC Davis and VA Palo Alto, respectively, for early next-day processing in the Debes Laboratory at TJU. Tissues from sIgM[−/−] and WT mice housed at TJU were harvested intact and stored overnight in complete RPMI at 4 °C for parallel analysis with knockout tissues.

### Cell isolation, stimulation, culture, and reconstitution of sIgM production

Cells were isolated from mouse spleens, LNs, and newborn livers by grinding tissues with a syringe plunger through 70-μm and 40-μm cell strainers. BM was flushed from the femurs using a 25 G needle. Red

blood cells were lysed from spleen, newborn liver, and BM samples using red cell lysis buffer. Peritoneal cavity cells were obtained by peritoneal lavage with ~9 ml PBS. Single cells from all tissues were washed, counted by hemocytometer and resuspended in complete RPMI (RPMI 1640 with 25 mM HEPES buffer, 1× Glutamax, 100 U/mL penicillin, 100 μg/mL streptomycin, 10% newborn calf serum, 50 μM 2-mercaptoethanol).

To analyze cytokine production, cells (1–4 × 10$^6$ cells/ml) were stimulated for 4 hours with 10 ng/ml phorbol 12-myristate 13-acetate (Sigma-Aldrich P8139), 0.5 μg/ml ionomycin (Sigma-Aldrich I0634), and 10 μg/ml LPS (Sigma-Aldrich L4641) at 37 °C 5% CO$_2$ 13% O$_2$. After 2 h, 10 μg/ml brefeldin-A (Sigma-Aldrich B6542) and 2 μM monensin (BD 554724) were added, except for samples from IL-10$^{GFP}$ mice.

For in vitro B-cell culture, CD19 microbeads (Miltenyi 130-121-301) were used following manufacturer's instructions to purify B cells (>98%) from single-cell splenic preparations. B cells were plated at 2 × 10$^6$ cells/ml and left untouched or stimulated with soluble 5 μg/ml anti-CD40 (BioXcell BE0016-2) or 1 μg/ml LPS (Sigma-Aldrich L2137) for 3 days, and supernatant was frozen at −80 °C until analysis by ELISA.

For reconstitution of sIgM production, 2.8–5.6 × 10$^6$ WT or sIgM$^{-/-}$ donor PerC cells were injected intraperitoneally into 4.5–7 week–old sIgM$^{-/-}$IL-10$^{GFP}$ recipient mice. 5-8 weeks post transfer, recipient mice were analyzed.

## Flow cytometry and FACS sorting

To stain cells for flow cytometry, cells were pre-incubated with mouse anti-CD16/CD32 (2.4G2; Bio X Cell) and rat IgG (Jackson Immunoresearch Laboratories 012-000-003) to reduce non-specific staining and Live/Dead Fixable Aqua Dead Cell Stain (ThermoFisher L34966) to exclude dead cells from analysis. Cells were stained with anti-mouse monoclonal antibodies listed in Supplementary Table 1. All staining and wash steps were performed at 4 °C in PBS with 0.2% bovine serum albumin (BSA; Sigma-Aldrich A9418). IL-10$^{GFP}$ reporter samples were immediately acquired on flow cytometers, and samples from non-reporter mice were fixed with 2% paraformaldehyde (Sigma-Aldrich 158127). Intracellular IL-10 staining was performed on fixed cells using buffer containing 0.5% saponin (Sigma-Aldrich 47036). All samples were pregated on single, lymphocyte scatter, Live/Dead Aqua$^-$CD45$^+$ cells. Splenic and lymph node B cells were identified as CD19$^+$ distinguishing B-1 cells as B220$^{lo/neg}$CD43$^+$ with CD5$^+$ (B-1a) and CD5$^-$(B-1b) and B-2 cells as B220$^{hi}$CD43$^{neg}$ cells. B-2 cells were further distinguished as transitional stage 1 (T1): CD24$^{hi}$CD21$^{lo/neg}$; transitional stage 2 (T2): CD24$^{hi}$CD21$^{int}$; transitional stage 2-marginal zone precursor (T2-MZP): CD24$^{hi}$CD21$^{hi}$, CD23$^+$; marginal zone (MZ): CD24$^{hi}$CD21$^{hi}$CD23$^{neg}$; and follicular (FO): CD24$^{int}$CD21$^{int}$CD23$^+$IgD$^+$ (Supplementary Fig. 1a). BM Hardy Fraction cells were gated on singlets, Live/Dead (L/D) stain$^-$CD3$^-$F4/80$^-$NK1.1$^-$Gr-1$^-$B220$^+$. B220$^+$ BM cells were further subdivided into Hardy fractions: Fr. A: B220$^+$CD43$^+$IgM$^-$BP1$^-$CD24$^-$; Fr. B: B220$^+$CD43$^+$IgM$^-$BP1$^-$CD24$^+$; Fr. C-C': B220$^+$CD43$^+$IgM$^-$BP1$^+$CD24$^+$; Fr. D: B220$^+$CD43$^-$IgM$^-$IgD$^-$; Fr. E: B220$^+$CD43$^-$IgM$^+$IgD$^-$; and Fr. F: B220$^+$CD43$^-$IgM$^{+/-}$IgD$^+$ (Supplementary Fig. 1b). Breg phenotype was identified by expression of TIM1$^+$, CD1d$^{hi}$CD5$^+$, or CD9$^+$ among total splenic B cells (Supplementary Figs. 1c–e). Plasma cell (PC) identity was assessed by flow cytometry directly ex vivo on L/D stain$^-$CD45$^+$CD3$^-$F4/80$^-$IgD$^-$Blimp-1$^{GFP+}$ lymphocytes using sIgM$^{-/-}$ mice crossed with Blimp-1$^{GFP}$ reporter mice (Supplementary Fig. 1f). For non-B-cell analysis, we assessed frequency of IL-10$^+$ cells based on GFP expression in non-stimulated samples among lymphocyte subsets: pregated on singlets, lymphocyte scatter, LiveDead$^-$CD45$^+$CD19$^-$ then subgated for CD4$^+$ T cells (CD3$^+$NK1.1$^-$CD4$^+$CD8$^-$), CD8$^+$ T cells (CD3$^+$NK1.1$^-$CD4$^-$CD8$^+$), γδ T cells (CD3$^+$NK1.1$^-$γδ TCR$^+$), NK cells (CD3$^-$NK1.1$^+$), NKT cells (CD3$^+$NK1.1$^+$), and myeloid/granulocyte subsets: pregated on singlets, cell scatter, LiveDead$^-$CD45$^+$CD19$^-$ then subgated on macrophages (F4/80$^+$), neutrophils (F4/80$^-$CD11b$^+$Ly6G$^+$Ly6C$^+$), inflammatory

monocytes (iMOs; F4/80$^-$CD11b$^+$Ly6G$^-$Ly6C$^{hi}$), eosinophils (F4/80$^-$CD11b$^+$Ly6G$^-$Ly6C$^{lo}$), and dendritic cells (F4/80$^-$CD11c$^+$) (Supplementary Fig. 3a, b). All samples were acquired on a BD LSRFortessa, LSRII, FACSymphonyA5, or FACSCelesta in the Sidney Kimmel Cancer Center Flow Cytometry and Human Immune Monitoring Facility at TJU using Data-Interpolating Variational Analysis software (BD Biosciences, version 8.0.1), and data was analyzed with FlowJo software (Tree Star, version 10) and graphed using Graphpad Prism (version 9). Appropriate isotype antibodies or non-reporter mice were used to establish gates.

To FACS sort for single-cell RNAseq, total splenocytes from sIgM$^{-/-}$IL-10$^{GFP}$ mice were stimulated at 4×10$^6$ cells/ml with 10 ng/ml phorbol 12-myristate 13-acetate, 0.5 μg/ml ionomycin, and 10 μg/ml LPS for 4-h at 37 °C, 5% CO$_2$ 13% O$_2$. B cells were enriched using CD19 microbeads (Miltenyi 130-121-301) following the manufacturer's protocol. Cells were blocked with Fc block and stained for viability and B-cell subset markers, washed and sorted on a FACSAria II. Cells were gated on MZ B cells, FO B cells, and B-1 B cells as described above with additional use of CD5 expression to delineate B-1a B cells. Each subset was further sorted into IL-10$^+$ (GFP$^+$) and IL-10$^-$ (GFP$^-$) as shown in Supplementary Fig. 4. On each sorted subset, cell viability was confirmed using a Countess II FL Automated Cell Counter (ThermoFisher).

## ELISA, RNA isolation, and qPCR

For total IgM ELISAs, Nunc 96 well Multisorp plates (ThermoFisher 467340) were coated with goat-anti-mouse IgM (Bethyl Labs A90-101A) in 0.05 M carbonate-bicarbonate pH 9.6 buffer (ThermoFisher 28382 or Sigma-Aldrich SRE0034) and stored at 4 °C. On the day of usage, plates were washed with ELISA wash buffer (1× PBS 0.025% Tween-20) at the start and between each step. The plates were blocked with 2% BSA (Sigma-Aldrich A9418) in PBS for 1–2 hours at room temperature with light shaking. ELISA standard mouse IgM (ThermoFisher 39-50470-65) was prepared according to manufacturer's instructions. Samples, ELISA standards, and secondary antibodies were appropriately diluted in 2% BSA, run in duplicates and, at each step, incubated for 1 h at RT with light shaking. Goat-anti-mouse IgM-HRP (Bethyl Labs A90-101P) was used as the secondary antibody. For development, plates were incubated with Sure-blue 1-component peroxidase substrate (SeraCare 5120-0075) at room temperature in the dark for 5–7 minutes. The reaction was stopped with 0.18 M H$_2$SO$_4$ and absorbance read at 450 nm on a plate reader (Berthold, Tristar2 LB942, ICE program version 1.0.9.8). To detect IL-10 by ELISA, the Mouse IL-10 Duoset ELISA (R&D Systems DY417-05) was performed according to manufacturer protocol for 3-day-culture supernatant analysis or with the following modifications for serum analysis: The amount of sample plated was 200 μl and the incubation time was extended to 2.5 hours. For all ELISAs, results were calculated using a 4-parameter logistic regression from the standard curve on Graphpad Prism (version 9).

Total RNA was extracted from single-cell suspensions of spleen and subiliac LNs using RNeasy Mini kit (QIAGEN 74106) and reverse transcribed with a high-capacity cDNA reverse transcription kit (ThermoFisher Scientific 4368814) following the manufacturer's protocols. For qPCR analysis, the following TaqMan probes (Thermo-Fisher Scientific) were used: RPLP0, Mm00725448_s1; Gapdh, Mm99999915; IL-10, Mm01288386_m1; Ighm, Mm01718956_m1. All samples were run in duplicate with housekeeping gene controls (*RplpO* or *Gapdh*) on Applied Biosystems StepOnePlus Real-Time PCR System (StepOne Software version 2.2.2). qPCR data was analyzed using Microsoft Excel (version 16), data was analyzed using housekeeping control genes, and each individual sample was run in duplicate.

## Single-cell transcriptome library preparation, sequencing, and bioinformatics analysis

For V(D)J BCR analysis, FACS-sorted B cells were resuspended in media aiming to capture 5000 cells per sample. Samples were barcoded with

a 10x Chromium Controller to generate single-cell gel bead-in emulsions according to the manufacturer's instructions (10x Genomics Inc.). For 10x Genomics single-cell RNA-sequencing, we sequenced the transcriptomes of 30,000 cells captured from IL-10+ and IL-10− B-1a, MZ, and FO samples. For each sample, V(D)J libraries were constructed using Chromium Single-Cell 3' v3 reagent kit following manufacturer's instructions and sequenced on NextSeq 500 using paired-end chemistry (26 bp read 1 and 98 bp read 2) with coverage of a minimum of 10,000 reads per cell. Of the 30,000 input cells (5000 × 6 samples), we recovered 16,229 sequenced cells after filtering with a minimum depth of 17,000 read pairs per cell. Raw sequencing reads were processed using the Cell Ranger v5.0 pipeline from 10X Genomics. Sequenced BCL files were converted to fastq using Cellranger 'mkfastq' function, and reads were demultiplexed between the appropriate samples. For V(D)J BCR, cellranger 'vdj' function was used to generate annotated single-cell V(D)J sequences for identification of CDR3 sequence and the rearranged V(D)J gene. For annotations, we used GRCm38 version of the mouse genome. Clonotype file from CellRanger was further processed to perform a downstream analysis. First, clonotypes were filtered to remove any sequenced cell that expresses T-cell markers CD3G and CD3D. Further, we excluded cells expressing *Il10* transcript from IL-10− samples. Finally, clonotypes and related CDR3 sequences of IL-10+ and IL-10− B-1a, MZ, and FO samples were compared individually and in groups. The filtering and plotting were performed using R programming language. Diversity Indices were assessed using scRepertoire[73]. The Fastq files have been deposited in NCBI GEO under accession number GSE229063.

Due to an unknown technical issue, we were unable to detect the heavy chain in most of the analyzed cells even upon repeat library construction. We sequenced *Ighm* exon 1 of splenic B cells from WT and sIgM−/− mice (*Ighm* 3 F: ATAGCCCTGCTGCAGTTTCG and *Ighm* 3 R: TGGTGAAGCCAGATTCCACG) and found that the 10X enrichment primer binding sites are not mutated in sIgM−/− mice, ensuring that the primers recognize *Ighm* in sIgM−/− mice. The cell stimulation caused downregulation of *Ighm*; however, we found similar average Ct values for *Ighm*, *Gapdh*, and *RplpO*, indicating sufficient material present in the starting material for analysis. An additional sort of WT versus sIgM−/− total splenic B cells yielded heavy chain data for both WT and sIgM−/− B cells, indicating that the lack of heavy chain detection in our six samples (IL-10+ and IL-10− B-1a, MZ, and FO) was not due to the genetic modification that removed the secretory exon of Ighm.

## Statistics and reproducibility

We performed experiments based on previously published studies in our laboratory and included a total number of 6−27 per group (including independent repeat experiments) except for the VDJ single-cell analysis, which used 1 sIgM−/−IL-10GFP mouse analyzing 30,000 input cells from six populations (IL-10+ and IL-10− cells from B1a, MZ, and FO B cells subsets). ELISA and qPCR samples were run in duplicate for each mouse. In repertoire analysis, clonotypes were filtered to remove any sequenced cell that expressed T-cell markers CD3G or CD3D, and we excluded *Il10* transcript-positive cells from IL-10− samples. No other data were excluded from the analyses. All mice were grouped by genotype, and all control groups were matched in age and sex. For adoptive transfer experiments, recipient mice were randomized to donor genotype, and the investigators were not blinded to allocation during experiments and outcome assessment. Nonparametric two-tailed Mann−Whitney $U$ test and Kruskal−Wallis with Dunn's multiple comparisons post-test were used as indicated in each figure legend. All data points represent individual biological samples. *ns* = not significant and $p < 0.05$ was considered statistically significant. $*p < 0.05$; $**p < 0.01$; $***p < 0.001$; $****p < 0.0001$. GraphPad Prism version 9 (San Diego, CA) was used to perform all statistical analyses.

## Reporting summary

Further information on research design is available in the Nature Portfolio Reporting Summary linked to this article.

## Data availability

Source data generated in this study are provided in the Supplementary Information/Source Data file. The V(D)J RNA-sequencing data generated in this study have been deposited in NCBI GEO under accession number GSE229063 and annotated using reference genome GRCm38, and all VDJ processed data are provided in Supplementary Data 1. All other data are available in the article and its Supplementary Information files or from the corresponding author upon request. Source data are provided with this paper.

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

## Acknowledgements

We thank Samantha Measner, Jam Mizzchel Yu, Shaiku Jalloh, and Zheng Luo for excellent technical assistance and Kishore Alugupalli, Tim Manser, Sangwon Kim, Neda Nikbakht, Dave Allman, and Michael Cancro for helpful discussions. We are indebted to the SKCC Flow Cytometry and Cell Sorting Facility (P30 CA056036) and TJU animal care staff and veterinarians for reliable support. The authors were supported in part by NIH grants AI130471 and AI047822 (E.C.B.), AI148652 and AI151995 (N.B.), AR067751, AI127389, and AI165750 (G.F.D.) and an Ethel Brown Foerderer Foundation Fellowship from Thomas Jefferson University and AI134646 (S.E.M.).

## Author contributions

S.E.M., L.E.A., and G.F.D. designed research; S.E.M., L.E.A., G.K., and R.B. performed research; G.F.D., E.C.B., and N.B. supervised research; E.C.B., N.B., and G.K. contributed key research resources or methodology; S.E.M., L.E.A., G.K., and G.F.D. analyzed data; S.E.M. and G.F.D. drafted the manuscript; S.E.M., L.E.A., G.K., R.B., E.C.B., N.B., and G.F.D. reviewed and edited the manuscript.

## Competing interests

The authors declare no competing interests.
