## [Peer Review File · Nature Communications]

Secreted IgM modulates IL-10 expression in B cellsEditorial Note: Parts of this Peer Review File have been redacted as indicated to maintain the confidentiality of unpublished data.

REVIEWER COMMENTS

Reviewer #1 (Remarks to the Author):

McGettigan et al. report an interesting and novel relationship between secreted IgM and IL-10 producing B cells. Using mice that lack the ability to secrete IgM, they demonstrate increased IL-10⁺ B cells (mRNA and protein expression) and increased titres of serum IL-10 in secreted IgM^{-/-} mice compared to wild-type. Clever experiments using heterozygous mice with B cells of IgMa (secreted IgM^{-/-}) and IgMb (secreted IgM^{+/+}) alleles showed that secreted IgM^{-/-} B cells did not increase IL-10 expression when in environment with secreted IgM, thereby confirming that the expansion of IL-10⁺ B cells was due to a lack of secreted IgM rather than an effect of the B cell's inability to secrete IgM itself.

The mechanism by which secreted IgM is regulating IL-10⁺ B cells is not completely elucidated as knockout of FcμR (total and conditional B cell) only partially increases IL-10 expression. However, these findings are well addressed in the discussion with extended description of other receptors likely to be involved in this pathway. Finally, there is an excellent discussion on the possible physiological roles of this IgM and IL-10 relationship in neonatal development and tissue.

Overall, this was a very well written and clear manuscript with an excellent experimental approach and analysis. The data supports the conclusions and these findings are novel and warrant publication.

Specific point to address:

1. Given that splenic CD4 T cells show a minor but significant increase in IL-10 expression, were any other cells evaluated? Please show the data for IL-10 expression in other cell types.

Reviewer #2 (Remarks to the Author):

Major Comments:

Dr. McGettigan et al. (in the laboratory of Dr. Debes at Thomas Jefferson University) describe that: in comparison between mice unable to secrete IgM due to ablation of its secretory exon (sIgM KO) and their WT controls, (i) the intracellular production of IL-10 by B cells in spleen and lymph nodes after brief stimulation with PMA/ionomycin/LPS (P/I/L) and the serum IL-10 levels were significantly increased in the mutant mice and (ii) this increase was observed with all splenic B cell subsets (i.e., B-1, T1, T2, T2-MZP, MZ, FO), including Breg cells defined by various markers, and occurred from birth to adulthood. The P/I/L-induced IL-10⁺ B cells in mice with either general or B cell-specific deletion of Fcμr (Fcμr KO) were intermediate between the WT and sIgM KO mice in terms of their frequencies and cell numbers. The authors propose that secreted IgM modulates IL-10 expression by B cells through the FcμR, but not other IgM-binding receptors (e.g., CD22, pIgR). Analysis of IL-10 expression by B cells in sIgM KO mice was performed extensively, including subpopulations, ontogeny (newborn to adult) and clonality, according to the known characteristics of IL-10⁺ Breg cells. These findings might be informative because there are few reports on this topic in sIgM KO mice. However, this reviewer has less enthusiasm for endorsement to publish the manuscript in its present form, because it is mostly descriptive and lacks any mechanistic view, especially how FcμR is involved in IL-10 expression in conjunction with the IgM BCR and secreted IgM.

Other Comments:

1. Of three different Cd22 KO mice [i.e., ref. 62 (this study); Immunity 5:551, 1996; Science 274:798, 1996], apparent discrepancies have been reported for some aspects. The authors could not reproduce the elevated IL-10 expression by their mutant B cells as described in ref. 9. Other than the usual causes of phenotype differences, e.g., strain differences, targeting exons and environments, if splenic B cells from the authors' Cd22 KO and WT mice are stimulated for a longer period (e.g., LPS for 48 hrs and P/I for the last 5 hrs) as done in ref. 9, do such activated B cells from the mutant mice produce more IL-10 than WT controls?

2. Likewise, in two different sIgM KO mice [ref. 24 (this study); PNAS 95:10089, 1998] IL-10 production by their B cells appears different. In the latter sIgM KO mice, more IL-10 secretion was

observed with mutant splenic B cells cultured for 3 days without stimulation or with agonistic anti-CD40 mAb, compared with WT B cells (J. Immunol. 186:4967, 2011). However, comparable IL-10 production in both groups of mice was observed with splenic B cells, either stimulated *ex vivo* with LPS for 3 days or isolated from mice one week after antigen-induced arthritis, suggesting that different stimuli resulted in different outcomes. If splenic B cells from the authors' sIgM KO and WT mice are stimulated with LPS for 3 days, do mutant B cells still produce more IL-10 than WT B cells?

3. It is also known that there are conflicting views on the phenotypes of the five different strains of FcμR KO mice, as reported in ref. 20. Splenic B cells from one of the mutant mice secreted much less IL-10 but comparable amounts of IL-6 upon stimulation with LPS for 2 days compared to WT B cells (Curr. Top. Microbiol. Immunol. 408:25, 2017). If splenic B cells from the authors' FcμR KO mice are stimulated in the same way, do they secrete more or less IL-10?

4. According to recent crystallographic and cryo-EM analyses, the two Ig-like domains of FcμR may interact with both sides of the C_μ4 domain of the membrane-bound, monomeric IgM BCR (i.e., a 2:1 stoichiometry for FcμR:IgM monomer) (Nature 615:907, 2023). Thus, if this is true, even in mutant mice lacking secreted IgM, FcμR may interact with the IgM BCR.

5. Other than CD22 and pIgR, another potentially important pentameric IgM-binding protein, called apoptosis inhibitor of macrophage (AIM), CD5 like antigen (CD5L) or soluble protein α (Spα), should also be considered in the context of this study.

6. If the authors employ acute (e.g., influenza virus) or chronic (e.g., Salmonella) pathogen infection and autoimmune (e.g., MOG-induced EAE) models, as described in refs. 61 and 66 herein, then the outcomes of these studies should provide more insights into the pathophysiological mechanisms of the interactions among IgM, FcμR and IL-10 and will attract more readers to Nature Communications.

Reviewer #3 (Remarks to the Author):

In this study the authors investigate the impact of secreted IgM on the frequency and diversity of IL-10⁺B cells in mice, with heavy reliance on the use of sIgM^{-/-} mice. Whereas IL-10⁺B cells are relevant in the context of Bregulatory (Breg) cells and their role in homeostasis and disease, the factors that govern their development are not determined, hence this topic is of interest.

The basis for the study emerged when the authors noticed unexpectedly that sIgM^{-/-} have increased levels of serum IL-10. Upon further analysis they identified an increased frequency of IL-10⁺ B cells in these mice after *in vitro* stimulation of B cells with PMA/ionomycin/LPS. The authors go on to conduct a systematic analysis of IL-10⁺B cell phenotype, proportion, and distribution in lymphoid tissues in wild type versus sIgM^{-/-} mice.

The main findings are straightforward, largely showing the increase in IL-10⁺ B cells extends across lymphoid compartments and across B cell subsets including several phenotypes previously associated with Breg. They do not appear to represent an oligoclonal expansion since BCR repertoire analysis of lambda light chain sequences show a breadth of clonotypes indicative of polyclonal populations across B cell subsets. Ontogeny studies show inverse correlations between sIgM and IL-10⁺B cells during development in wildtype and sIgM^{-/-} mice. Finally, IL-10⁺B cell proportions are partially impacted by the absence of FcμR as demonstrated in FcμR^{-/-} mice suggesting this molecule has a role in regulation of IL-10 expression.

In general the studies are well performed and presented but leave some questions remaining that could provide more mechanistic insight into the observations.

Questions:

1. These studies seem to primarily pertain to B cell development and homeostasis, but it would be of interest to determine the relationship of sIgM and IL-10⁺B cells in the setting of intentional immunization, infection, or chronic disease model. Can the authors test this in their sIgM^{-/-} model?

2. The authors note they also see a significant increase in splenic IL-10+ CD4 T cells as noted in supplementary data. Can they elaborate on this finding and provide insights into the relevance to the general B cell observations they report?

3. The data in Figure 5 implicates the Fc μ R receptor in regulation of IL-10 in B cells. The authors should comment or show the extent of expression of Fc μ R on the various B cell subsets as well as CD4+ T cells that they have shown are impacted by the absence of sIgM?

4. The data presented largely identify correlations and are sometimes overinterpreted. For example in Fig 4 the findings show correlations between IL-10+ B cells and sIgM levels. This does not seem to warrant the statement on page 9, line 9 that ‘..the presence of sIgM from birth on limits expansion of the IL0-10+ B cell pool’.

5. Can the authors test this relationship between sIgM and development of IL-10+ B cells more directly in vitro or in vivo by addition of exogenous sIgM to evaluate the impact on IL-10+ B cell proportions?

6. The observed inverse relationship between sIgM and IL-10+ B cells is established but not mechanistically linked except for a partial effect via loss of Fc μ R raising the question of how sIgM actually impacts IL-10+ B cell development.

Point-by-point-reply to Reviewer concerns

The authors thank the reviewers and editors for their thorough reviews and thoughtful comments. In the revised manuscript, we have addressed all concerns raised and included exciting new data.

Reviewer #1 (Remarks to the Author):

1. Given that splenic CD4 T cells show a minor but significant increase in IL-10 expression, were any other cells evaluated? Please show the data for IL-10 expression in other cell types.

To address the reviewer's comment, we included new data comparing IL-10 expression in CD4⁺ T cells, CD8⁺ T cells, NK T cells, $\gamma\delta$ T cells, NK cells, macrophages, dendritic cells, inflammatory monocytes, neutrophils, and eosinophils. We did not detect a significant increase in IL-10⁺ cells in any of the listed cell types in sIgM^{-/-} compared to WT mice except for CD4⁺ T cells whose increase in IL-10⁺ is minor relative to the phenotype observed in B cells. It is described that IL-10⁺ B cells may drive development of IL-10⁺ CD4⁺ T cells and could represent one reason for the isolated increase in IL-10⁺ CD4⁺ T cells. We have included a new Figure, and updated the text (new Supplementary Figure 2C; P6: L1-5)

Reviewer #2 (Remarks to the Author):

Major Comments:

Dr. McGettigan et al. (in the laboratory of Dr. Debes at Thomas Jefferson University) describe that: in comparison between mice unable to secrete IgM due to ablation of its secretory exon (sIgM KO) and their WT controls, (i) the intracellular production of IL-10 by B cells in spleen and lymph nodes after brief stimulation with PMA/ionomycin/LPS (P/I/L) and the serum IL-10 levels were significantly increased in the mutant mice and (ii) this increase was observed with all splenic B cell subsets (i.e., B-1, T1, T2, T2-MZP, MZ, FO), including Breg cells defined by various markers, and occurred from birth to adulthood. The P/I/L-induced IL-10⁺ B cells in mice with either general or B cell-specific deletion of Fc μ R (Fc μ R KO) were intermediate between the WT and sIgM KO mice in terms of their frequencies and cell numbers. The authors propose that secreted IgM modulates IL-10 expression by B cells through the Fc μ R, but not other IgM-binding receptors (e.g., CD22, pIgR).

Analysis of IL-10 expression by B cells in sIgM KO mice was performed extensively, including subpopulations, ontogeny (newborn to adult) and clonality, according to the known characteristics of IL-10⁺ Breg cells. These findings might be informative because there are few reports on this topic in sIgM KO mice. However, this reviewer has less enthusiasm for endorsement to publish the manuscript in its present form, because it is mostly descriptive and lacks any mechanistic view, especially how Fc μ R is involved in IL-10 expression in conjunction with the IgM BCR and secreted IgM.

We thank the reviewer for the constructive feedback. Our manuscript establishes a novel relationship between secreted IgM in the environment and IL-10 expression programming in B cells during ontogeny, which includes mechanistic experiments to solidify the phenotype and relationship. In the revised manuscript, we have added experiments that partially restore sIgM production in sIgM^{-/-} mice and lead to a significant reduction in IL-10⁺ B cells, validating our conclusion that sIgM regulates IL-10 expression programming in B cells (new Figure 4i). Extensive studies regarding the specific receptors beyond Fc μ R as well as receptor signaling

details are important but outside of the scope of the current manuscript. Importantly, most mouse strains necessary to perform such studies are not readily and/or commercially available (see below) limiting the ability to perform the proposed experiments in a realistic time frame. However, to address the concern, we have included a more extensive discussion on the relationship between sIgM and sIgM-binding receptors as well as the potential role of AIM/CD5L. (P14: L17-20, 22-23; P15: L1-5, L9-12, 17-20; P16: L8-16). Please also see below for further details.

Other Comments:

1. Of three different Cd22 KO mice [i.e., ref. 62 (this study); *Immunity* 5:551, 1996; *Science* 274:798, 1996], apparent discrepancies have been reported for some aspects. The authors could not reproduce the elevated IL-10 expression by their mutant B cells as described in ref. 9. Other than the usual causes of phenotype differences, e.g., strain differences, targeting exons and environments, if splenic B cells from the authors' Cd22 KO and WT mice are stimulated for a longer period (e.g., LPS for 48 hrs and P/I for the last 5 hrs) as done in ref. 9, do such activated B cells from the mutant mice produce more IL-10 than WT controls?

Unfortunately, we were unable to perform the requested experiments because we lost access to CD22-deficient mice. These mice are not readily commercially available (only as cryopreserved embryos). However, CD22 is only one of many sIgM binding receptors and our study will never be able to address all potential discrepancies between different knockout mice for CD22 and the potential shortcomings of each strain. Therefore, the reviewer is correct that we cannot formally exclude CD22 as a key participant in the regulation of sIgM-driven IL-10 programming in B cells. To reflect this possibility, we have updated our Discussion (P15: L9-12).

2. Likewise, in two different sIgM KO mice [ref. 24 (this study); *PNAS* 95:10089, 1998] IL-10 production by their B cells appears different. In the latter sIgM KO mice, more IL-10 secretion was observed with mutant splenic B cells cultured for 3 days without stimulation or with agonistic anti-CD40 mAb, compared with WT B cells (*J. Immunol.* 186:4967, 2011). However, comparable IL-10 production in both groups of mice was observed with splenic B cells, either stimulated *ex vivo* with LPS for 3 days or isolated from mice one week after antigen-induced arthritis, suggesting that different stimuli resulted in different outcomes. If splenic B cells from the authors' sIgM KO and WT mice are stimulated with LPS for 3 days, do mutant B cells still produce more IL-10 than WT B cells?

As pointed out correctly by the reviewer, the Neuberger sIgM^{-/-} strain (*Ehrenstein et al. 1998, PNAS 95:10089*) has a similar *ex vivo* baseline increase in IL-10 production in B cells (*Notley et al. 2011, J. Immunol. 186:4967; Fig. 3C*) as we report here for the Chen strain (*Boes et al. 1998, J. Immunol 160:4776*) that we fully backcrossed to C57BL/6 for our studies. As suggested, we performed 3-day stimulation of sIgM^{-/-} B cells with LPS or anti-CD40, and we detected a significant increase in IL-10 production in sIgM^{-/-} B cells relative to WT under both conditions in agreement with the baseline and anti-CD40 results published by Notley et al. It is unclear why the Notley study only detected an increase in IL-10 expression by sIgM^{-/-} B cells at baseline and after stimulation with anti-CD40 but not with LPS – for example, the quality of their LPS might have been compromised. To clarify this would require a true side-by-side comparison with B

cells from the two mouse strains. However, we have added our new data on different B cell stimulation conditions to the manuscript (new Supplementary Fig. 2a; P5: L16-19)

3. It is also known that there are conflicting views on the phenotypes of the five different strains of FcμR KO mice, as reported in ref. 20. Splenic B cells from one of the mutant mice secreted much less IL-10 but comparable amounts of IL-6 upon stimulation with LPS for 2 days compared to WT B cells (*Curr. Top. Microbiol. Immunol.* 408:25, 2017). If splenic B cells from the authors' FcμR KO mice are stimulated in the same way, do they secrete more or less IL-10?

The authors agree with the concerns about the discrepancies in phenotype between the five FcμR-deficient mouse strains (discussed extensively in *Kubagawa et al., 2019, Front. Immunol.* 10:945). The phenotype of FcμR-deficient mice with regard to IL-10 production in B cells in our study is consistent with that of the FcμR^{-/-} mice generated by Lee et al. (*Yu et al. 2018, J. Clin. Invest.* 128:1820). It is unclear why the Kubagawa et al. review article (*Kubagawa et al., 2017, Curr. Top. Microbiol. Immunol.* 408:25) did not report the same findings. However, as typical for a review article, experimental details of their B cell stimulation conditions were not described. It is possible that differences in the genetic targeting of the different strains affect IL-10 production because mouse *Il10* and *Fcμr* gene loci are in close proximity (~139 kb apart) on chromosome 1. Interestingly, the same review article by Kubagawa et al. states regarding potential differences in IL-10 production between the Ohno and Mak strains of FcμR-KO mice: “A side-by-side analysis of these two different strains of *Fcμr/Toso* KO mice would facilitate the resolution of these conflicting results, and it is highly likely that this discrepancy results from different strategies for gene targeting (deletion of exon 2-4 without *Neo* for ours versus deletion of exon 2-8 with remaining of *Neo* in the mouse genome for Brenner et al.)” (*Kubagawa et al., 2017, Curr. Top. Microbiol. Immunol.* 408:25, p.39) We agree with the notion of the Kubagawa et al. review article and a comprehensive study of the various FcμR-KO strains would be required to resolve potential differences in IL-10 expression by B cells. The strain in our study (*Nguyen et al. 2017, Nat. Imm.* 18:321) only targeted exon 4, whereas the other FcμR-KO strains targeted multiple exons which may contribute to potential differences in phenotype. To address the concerns, we have now included Discussion of this topic (P14: L22-23; P15: L1-5).

4. According to recent crystallographic and cryo-EM analyses, the two Ig-like domains of FcμR may interact with both sides of the Cμ4 domain of the membrane-bound, monomeric IgM BCR (i.e., a 2:1 stoichiometry for FcμR:IgM monomer) (*Nature* 615:907, 2023). Thus, if this is true, even in mutant mice lacking secreted IgM, FcμR may interact with the IgM BCR.

We are grateful for the suggestion to discuss this recent study (*Li et al. 2023, Nature* 615:907) that was published after submission of our manuscript. An additional study of similar focus (*Chen et al. 2023, Nat. Struct. Mol. Biol.* 30:1033) and commentary (*Sutton, 2023, Nat. Struct. Mol. Biol.* 30:866) were published shortly after. Together, these papers demonstrate using cryo-EM of FcμR that one pentameric sIgM has 8 potential binding sites for FcμR with demonstrated binding of 4 FcμR molecules and that one BCR IgM binds two FcμR molecules. The signaling capacities of FcμR are largely unexplored. Potential for clustering of FcμR, conserved tyrosine/serine residues in the cytoplasmic tail of FcμR, or interactions via putative adaptor proteins could modulate BCR signaling (positively or negatively) (*Kubagawa et al., 2019, Front. Immunol.* 10:945; *Sutton, 2023, Nat. Struct. Mol. Biol.* 30:866). Furthermore, FcμR binding to

the IgM-BCR in the absence of sIgM is possible and variations in BCR signaling strength and subsequent IL-10 programming could potentially occur dependent on sIgM presence or absence. Together, these papers on multimeric Fc μ R and IgM interactions support a mechanistic avenue for sIgM affecting B-cell IL-10 expression programming via modulating BCR signaling through Fc μ R. We have included these exciting findings in our Discussion (P16: L8-16).

5. Other than CD22 and pIgR, another potentially important pentameric IgM-binding protein, called apoptosis inhibitor of macrophage (AIM), CD5 like antigen (CD5L) or soluble protein α (Sp α), should also be considered in the context of this study.

As requested, we have now added discussion of AIM/CD5L/Sp α to the manuscript (P15: L17-20).

6. If the authors employ acute (e.g., influenza virus) or chronic (e.g., Salmonella) pathogen infection and autoimmune (e.g., MOG-induced EAE) models, as described in refs. 61 and 66 herein, then the outcomes of these studies should provide more insights into the pathophysiologic mechanisms of the interactions among IgM, Fc μ R and IL-10 and will attract more readers to Nature Communications.

This is an important question raised by the reviewer. As suggested, we have performed experiments with the goal to determine the inflammatory phenotype of sIgM^{-/-} mice. We were not able to perform the suggested infection or autoimmune models as we do not have local collaborators available for these models. Additionally, sIgM^{-/-} mice on fully backcrossed B6 background are not commercially available, and transfer to external collaborators would require a minimum of 9 months due to MTA agreements and quarantines.

[REDACTED]

[REDACTED]

However, as of today, we have not been successful in generating enough mice to perform these experiments. Therefore, performing these additional experiments will take a significant amount of time and further delay publication of our current manuscript without substantially contributing to its main focus on the role of sIgM in

regulating IL-10 expression programming in B cells. We have updated the text to indicate that future studies should be performed to address the role of enhanced B cell IL-10 expression in limiting immune responses (P17: L7-9).

Reviewer #3 (Remarks to the Author):

In this study the authors investigate the impact of secreted IgM on the frequency and diversity of IL-10+ B cells in mice, with heavy reliance on the use of sIgM^{-/-} mice. Whereas IL-10+ B cells are relevant in the context of B regulatory (Breg) cells and their role in homeostasis and disease, the factors that govern their development are not determined, hence this topic is of interest.

The basis for the study emerged when the authors noticed unexpectedly that sIgM^{-/-} have increased levels of serum IL-10. Upon further analysis they identified an increased frequency of IL-10+ B cells in these mice after in vitro stimulation of B cells with PMA/ionomycin/LPS. The authors go on to conduct a systematic analysis of IL-10+ B cell phenotype, proportion, and distribution in lymphoid tissues in wild type versus sIgM^{-/-} mice.

The main findings are straightforward, largely showing the increase in IL-10+ B cells extends across lymphoid compartments and across B cell subsets including several phenotypes previously associated with Breg. They do not appear to represent an oligoclonal expansion since BCR repertoire analysis of lambda light chain sequences show a breadth of clonotypes indicative of polyclonal populations across B cell subsets. Ontogeny studies show inverse correlations between sIgM and IL-10+ B cells during development in wildtype and sIgM^{-/-} mice. Finally, IL-10+ B cell proportions are partially impacted by the absence of FcμR as demonstrated in FcμR^{-/-} mice suggesting this molecule has a role in regulation of IL-10 expression.

In general the studies are well performed and presented but leave some questions remaining that could provide more mechanistic insight into the observations.

Questions:

1. These studies seem to primarily pertain to B cell development and homeostasis, but it would be of interest to determine the relationship of sIgM and IL-10+B cells in the setting of intentional Immunization, infection, or chronic disease model. Can the authors test this in their sIgM^{-/-} model?

The authors agree that this is an interesting question and we have performed extensive experiments to address this point. See above (response to Reviewer 2 #6).

2. The authors note they also see a significant increase in splenic IL-10+ CD4 T cells as noted in supplementary data. Can they elaborate on this finding and provide insights into the relevance to the general B cell observations they report?

To address this valid point, also explained above for Reviewer 1, we included new data comparing IL-10 expression in CD4⁺ T cells, CD8⁺ T cells, NK T cells, γδ T cells, NK cells, macrophages, dendritic cells, inflammatory monocytes, neutrophils, and eosinophils. We did not

detect a significant increase in IL-10⁺ cells in any of the listed cell types in sIgM^{-/-} compared to WT mice except for CD4⁺ T cells whose increase in IL-10⁺ is minor relative to the phenotype observed in B cells. It is described that IL-10⁺ B cells may drive development of IL-10⁺ CD4⁺ T cells and this could represent one reason for the isolated increase in IL-10⁺ CD4⁺ T cells. We have included a new Figure and updated the text (new Supplementary Figure 2C; P6: L1-5)

3. The data in Figure 5 implicates the FcμR receptor in regulation of IL-10 in B cells. The authors should comment or show the extent of expression of FcμR on the various B cell subsets as well as CD4⁺ T cells that they have shown are impacted by the absence of sIgM?

We thank the reviewer for raising this important point. FcμR is expressed on all B cell subsets (*Kubagawa et al. 2019, Front. Immunol. 10:945*), in line with our data showing increase in IL-10 expression among all major B cell subsets in sIgM^{-/-} mice. Strikingly, FcμR expression is highest on the T2-MZP and FO subsets (*Liu et al. 2019, Front. Immunol. 10:529*), consistent with our data demonstrating the highest relative increase in IL-10 expression in FO and T2-MZP B cells in sIgM^{-/-} mice. Specifically, the highest average fold-increase in percent IL-10⁺ B cells was in the FO (8.8-fold), followed by T2-MZP (7.0-fold), T2 (5.7-fold), T1 (4.1-fold), MZ (4.0-fold), and B-1 (1.2-fold) B cell subsets (Fig. 2a; p<0.001). Thus, the data support a major role for B cell-expressed FcμR in sIgM-driven regulation of IL-10 competency in B cells. Of note murine CD4⁺ T cells do not express FcμR, which excludes its direct function in IL-10 induction in CD4⁺ T cells (see also comment #2). We have updated text to include details on FcμR on CD4⁺ T cells and fold change in IL-10⁺ B cells by subsets (P6: L2-4, L14-17; P14: L17-20).

4. The data presented largely identify correlations and are sometimes overinterpreted. For example in Fig 4 the findings show correlations between IL-10⁺ B cells and sIgM levels. This does not seem to warrant the statement on page 9, line 9 that ‘..the presence of sIgM from birth on limits expansion of the IL-10⁺ B cell pool’.

To address the reviewer’s concern and to strengthen our conclusions, we have performed new experiments. Specifically, we have added experiments that restore sIgM production in sIgM^{-/-} mice by adapting a system by Reynolds *et al.*, in which transfer of WT peritoneal cavity (PerC) cells leads to reconstitution of IgM secreting cells in BM of recipients (*Reynolds et al. 2015, J. Immunol 194:231*). The restoration of sIgM production in sIgM^{-/-} mice significantly reduced the IL-10⁺ B cell fraction and supports our statement that the presence of sIgM limits expansion of the IL-10⁺ B cell pool. (See Point 5 for further details). We now added new data and revised the text (new Figure 4i; P10: L20-23; P11: L1-4). In addition, we have rephrased other statements (P11: L18-19; P14: L22-23; P15: L1-5, L9-12)

5. Can the authors test this relationship between sIgM and development of IL-10⁺ B cells more directly in vitro or in vivo by addition of exogenous sIgM to evaluate the impact on IL-10⁺ B cell proportions?

We agree with the Reviewer that this is an important question. Given the short half-life of sIgM in vivo (~2 days), we chose to reconstitute sIgM production rather than performing repeated soluble IgM injections. Specifically, we adapted the system by Reynolds *et al.*, in which donor PerC cells home into the BM and spleen and subsequently secrete sIgM (*Reynolds et al. 2015, J.*

Immunol 194:231). We adoptively transferred PerC cells from WT or sIgM^{-/-} donor mice into 4.5-7-week-old sIgM^{-/-}IL10^{GFP} recipient mice. Reconstitution of sIgM^{-/-} mice with WT PerC cells resulted in ~4.4 µg/ml serum sIgM, the equivalent of ~3% age-matched WT serum. Even this low relative amount of sIgM production (evaluated as serum sIgM) significantly reduced the frequency of IL-10⁺ cells among total B cells, B-1, MZ, and FO B cells by 15-40% relative to untreated sIgM^{-/-} mice (p<0.01). In contrast, reconstitution of sIgM^{-/-} mice with sIgM^{-/-} PerC cells did not induce serum IgM or alter frequencies of IL-10⁺ B cells. These data were included in the revised manuscript (new Figure 4i; P10: L20-23; P11: L1-4; P12: L3-6).

6. The observed inverse relationship between sIgM and IL-10⁺ B cells is established but not mechanistically linked except for a partial effect via loss of FcµR raising the question of how sIgM actually impacts IL-10⁺ B cell development.

This is an important question raised by the reviewer. Recent reports on the interactions of FcµR binding to sIgM and IgM BCR support a role in modulating BCR signaling. One pentameric sIgM has 8 potential binding sites for FcµR with demonstrated binding of 4 FcµR molecules and one BCR IgM binds two FcµR (*Chen et al. 2023, Nat. Struct. Mol. Biol. 30:1033; Li et al. 2023, Nature 615:907; Sutton, 2023, Nat. Struct. Mol. Biol. 30:866*). The signaling capacities of FcµR are largely unexplored. Potential for clustering of FcµR, conserved tyrosine/serine residues in the cytoplasmic tail of FcµR, or interactions via putative adaptor proteins could modulate BCR signaling (positively or negatively) (*Kubagawa et al. 2019, Front. Immunol. 10:945; Sutton, 2023, Nat. Struct. Mol. Biol. 30:866*). Together, these papers on multimeric FcµR and IgM interactions support a mechanistic avenue for sIgM affecting B cell IL-10 expression programming via modulating BCR signaling through FcµR. We have included these exciting findings in our Discussion (P16: L8-16).

REVIEWERS' COMMENTS

Reviewer #1 (Remarks to the Author):

The authors have responded to my comments with the inclusion of additional supplementary data and have added key discussion points around a potential mechanism for their findings.

Reviewer #2 (Remarks to the Author):

This reviewer endorses to publish the revised manuscript NCOMMS-23-11063A.

Reviewer #3 (Remarks to the Author):

The authors have adequately addressed my concerns and the new data included have improved the manuscript.

Point-by-point reply to reviewers' comments

Reviewer #1 (Remarks to the Author):

The authors have responded to my comments with the inclusion of additional supplementary data and have added key discussion points around a potential mechanism for their findings.

Thank you for your assessment and time.

Reviewer #2 (Remarks to the Author):

This reviewer endorses to publish the revised manuscript NCOMMS-23-11063A.

Thank you for your assessment and time.

Reviewer #3 (Remarks to the Author):

The authors have adequately addressed my concerns and the new data included have improved the manuscript.

Thank you for your assessment and time.